# European anthropogenic AFOLU greenhouse gas emissions: a review and benchmark data

Ana Maria Roxana Petrescu[1], Glen P. Peters[2], Greet Janssens-Maenhout[3], Philippe Ciais[4], Francesco N. Tubiello[5], Giacomo Grassi[3], Gert-Jan Nabuurs[6], Adrian Leip[3], Gema Carmona-Garcia[3], Wilfried Winiwarter[7,8], Lena Höglund-Isaksson[7], Dirk Günther[9], Efisio Solazzo[3], Anja Kiesow[9], Ana Bastos[10], Julia Pongratz[10,11], Julia E.M.S. Nabel[11], Giulia Conchedda[5], Roberto Pilli[3], Robbie M. Andrew[2], Mart-Jan Schelhaas[6] and Han Dolman[1]

[1] Department of Earth Sciences, Vrije Universiteit Amsterdam, 1081HV, Amsterdam, The Netherlands
[2] CICERO Center for International Climate Research, Oslo, 0349, Norway
[3] European Commission, Joint Research Centre EC-JRC, Ispra, 21027, Italy
[4] Le Laboratoire des Sciences du Climat et de l'Environnement LSCE, Gif sur Yvette Cedex, F-91191, France
[5] Food and Agriculture Organization FAO, Statistics Division, Rome, 00153, Italy
[6] Wageningen University WUR, Wageningen, 6708PB, The Netherlands
[7] International Institute for Applied Systems Analysis IIASA, 2361 Laxenburg, Austria
[8] Institute of Environmental Engineering, University of Zielona Góra, Zielona Góra, PL 65-417, Poland
[9] Umweltbundesamt UBA, Berlin, 14193, Germany
[10] Departent of Geography, Ludwig-Maximilians-University of Munich, Munich, 80333, Germany
[11] Max Planck Institute for Meteorology, D-20146, Hamburg, Germany

*Correspondence to*: A.M. Roxana Petrescu (a.m.r.petrescu@vu.nl)

## Abstract

Emission of greenhouse gases (GHG) and removals from land, including both anthropogenic and natural fluxes, require reliable quantification, including estimates of uncertainties, to support credible mitigation action under the Paris Agreement. This study provides a state-of-the-art scientific overview of bottom-up anthropogenic emissions data from agriculture, forestry and other land use (AFOLU) in the European Union (EU28[1]). The data integrates recent AFOLU emission inventories with ecosystem data and land carbon models and summarizes GHG emissions and removals over the period 1990-2016. This compilation of bottom-up estimates of the AFOLU GHG emissions of European national greenhouse gas inventories (NGHGI) with those of land carbon models and observation-based estimates of large-scale GHG fluxes, aims at improving the overall estimates of the GHG balance in Europe with respect to land GHG emissions and removals. Whenever available, we present uncertainties, its propagation and role in the comparison of different estimates. While NGHGI data for EU28 provides consistent quantification of uncertainty following the established IPCC guidelines, uncertainty in the estimates produced with other methods needs to account for both within model uncertainty and the spread from different model results. The largest inconsistencies between EU28 estimates are mainly due to different sources of data related to human activity, referred here as activity data (AD) and methodologies (Tiers) used for calculating emissions and removals from AFOLU sectors. The

---

[1] We refer to EU28 as communicated by EUROSTAT, including UK: https://ec.europa.eu/eurostat/help/faq/brexit. As of 1 February 2020, the UK is no longer be a part of the European Union.

referenced datasets related to figures are visualised at https://zenodo.org/record/3662371#.Xkui-WhKjIU Petrescu et
al., 2020.

**How to cite:** Petrescu, A. M. R., Peters, G. P., Janssens-Maenhout, G., Ciais. P., Tubiello, F. N., Grassi, G., Nabuurs, G-J., Leip, A., Carmona-Garcia, G., Winiwarter, W., Höglund-Isaksson, L., Günther, D., Solazzo, E., Kiesow, A., Bastos, A., Pongratz, J., Nabel, J.E.M.S., Conchedda, G., Pilli, R., Andrew, R. M., Schelhaas, M-J. and Dolman, A. J.: European anthropogenic AFOLU greenhouse gas emissions: a review and benchmark data, Earth Syst. Sci. Data Discuss., essd-2019-199, in review, 2019.

## 1. Introduction

The atmospheric concentrations of the main greenhouse gases (GHG) have increased significantly since pre-industrial times (pre-1750), by 46 % for carbon dioxide ($CO_2$), 257 % for methane ($CH_4$) and 122 % for nitrous oxide ($N_2O$) (WMO 2019). The rise of $CO_2$ levels is caused primarily by fossil fuel combustion, with a substantial contributions from land use change. Increases in emissions of $CH_4$ are mainly driven by agriculture and by fossil fuel extraction activities, while increases in natural emissions post-2006 cannot be ruled out (e.g. Worden et al., 2017). Increases in $N_2O$ emissions are largely due to anthropogenic activities, mainly in relation to the application of nitrogen (N) fertilizers in agriculture (FAO 2015; IPCC SRCCL 2019). Globally, fossil fuel emissions grew at a rate of 1.5 % $yr^{-1}$ for the decade 2008–2017 and account for 87 % of the anthropogenic sources in the total carbon budget (Le Quéré et al., 2018). In contrast, global emissions from land use change were estimated from bookkeeping models and land carbon models (Dynamic Global Vegetation Models, DGVMs) to be approximately stable in the same period, albeit with large uncertainties (Le Quéré et al., 2018). Importantly, emissions arising from land management changes were not estimated in the global carbon budget.

National greenhouse gas inventories (NGHGI) are prepared and reported by countries based on IPCC Guidelines (GLs) using national data and different calculation methods (Tiers) for well-defined sectors. The IPCC tiers represent the level of sophistication used to estimate emissions, with Tier 1 based on default assumptions, Tier 2 similar to Tier 1 but based on country specific parameters, and Tier 3 based on the most detailed process-level estimates (i.e. models).

After 2020, European countries will report their GHG emissions reductions following the newly approved UNFCCC transparency framework (UNFCCC, 2018), including the reporting principles of Transparency, Accuracy, Completeness, Consistency and Comparability (TACCC), and using the IPCC methodological guidance (IPCC 2006 GLs). Furthermore, the IPCC 2019 Refinement (that may be used complementing the IPCC 2006 GLs) has updated guidance on the possible and voluntary use of atmospheric data for independent verification of GHG inventories. So far, only few countries (e.g. Switzerland, UK and Australia) are already using atmospheric GHG measurements, on a

voluntary basis, as an additional consistency check of their national inventories. Annex I[2] countries (including the EU)
submit annually complete inventories of GHG emissions from the 1990 base year[3] until two years before the current

reported year, and these inventories are all reviewed to ensure TACCC.  This allows for most of these Annex I
countries to track progress towards their reduction targets committed for the Kyoto Protocol (UNFCCC, 1997) and
now for the Paris Agreement (PA) (Paris Agreement, 2015).

According to UNFCCC 2018 NGHGI estimates, the European Union (EU28) emitted 3.9 Gt $CO_2$ equivalents
($CO_2$-eq) in 2016 (incl. LULUCF/FOLU[4]) and 4.2 Gt $CO_2$-eq (excl. LULUCF) (the GWP100 metrics[5] (IPCC, 2007)

is here used to compare different gases in $CO_2$-eq). These anthropogenic emissions, incl. LULUCF, represent about 8
% of the world total. This number is consistent with the EDGAR v4.3.2FT2017 inventory (Olivier and Peters, 2018)
using IEA (2017) and BP (2018) data for energy sectors and EDGARv4.3.2 (Janssens-Maenhout et al., 2019) and
FAOSTAT 2018 for other (mainly agricultural and land use) sectors. A few large economies accounted for the largest
share of EU28 emissions, with UK and Germany representing 33 % of the total EU28 emissions.

According to NGHGI 2018 data, total anthropogenic emission of GHGs  in the EU28 (Fig. 1) decreased by
24 % from 1990 to 2016 (UNFCCC, 2018). $CO_2$ emissions (incl. LULUCF) account for 81 % of the total EU28
emissions  in 2016, and declined 24 % since 1990, accounting for 71 % of the total reduction in GHG emissions. $CH_4$
emissions account for 10 % of and $N_2O$ for 19 % of total GHG emissions, both gases have had  a reduction of 37 %
from 1990 levels. These reductions were due to both European and country specific policies on agriculture and

environment implemented in the early 1990s (e.g., the nitrogen directive which limited the amount of N use in
agriculture with repercussions to both fertilizer use and livestock numbers) and energy policies in the 2000s, (e.g. the
EU Emissions Trading System (ETS), and support for renewable energy and energy efficiency). The specific policies
triggered lower levels of mining activities, smaller livestock numbers, as well as lower emissions from managed waste
disposal on land and from agricultural soils. Specific historical structural changes in the economy linked to the collapse

of eastern European economies in early 1990's, the discovery and development of large natural gas sources in the
North Sea, and more recently the economic recession in 2009-2012, contributed as well to these diminishing trends
(Karstensen et al 2018). A few large, populous countries account for the largest share of EU28 emissions (UK and
Germany combined represent 33 % of the total) while the reduction of total emissions in 2016 compared to 1990 is
led by UK (38 %), Germany (24 %), Spain (23 %) , Poland (18 %), Italy (15 %) and France (11 %) (Olivier and Peters,

2018).

---

[2] Annex I Parties include the industrialized countries that were members of the OECD (Organization for Economic Co-operation and Development) in 1992, plus countries with economies in transition (the EIT Parties), including the Russian Federation, the Baltic States, and several Central and Eastern European States (UNFCCC, https://unfccc.int/parties-observers).

[3] For most Annex I Parties, the historical base year is 1990. However, Parties included in Annex I with an Economy in Transition during the early 1990s  (EIT Parties) were allowed to choose one year up to few years  before 1990 as reference because of a non-representative collapse during the breakup of the Soviet Union (e.g. Bulgaria (1988), Hungary (1985-87), Poland (1988), Romania (1989) and Slovenia (1986).

[4] In this study we refer to LULUCF (Land use, Land use Change and Forestry) which is the same as FOLU (Forestry and Other land use). The FOLU naming is mostly used in combination to Agriculture (AFOLU) since mitigation of GHG potential and efforts are focused on both sectors and represents a new sector in IPCC AR5, while countries in NGHGI report $CO_2$ from the LULUCF sector. It may be confusing using terminology such as: incl./excl. FOLU while incl./excl. LULUCF is widely used.

[5] GWP100 refers to the Global Warming Potential for 100 year time horizon. Under UNFCCC reporting and SBSTA 34 (2011), GWPs are a well-defined metric based on radiative forcing that continues to be useful in a multigas approach. NGHGI UNFCCC 2018 submissions use AR4 IPCC report as scientific base for GWP conversion factors ($CH_4$ = 25 and $N_2O$ = 298)

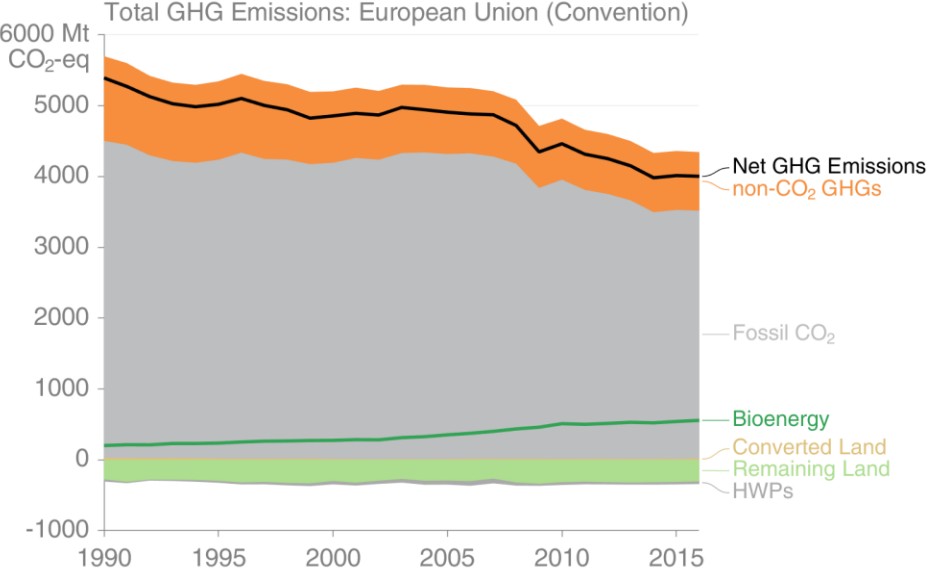

*Figure 1: Total reported EU28 GHG emissions according to UNFCCC NGHGI 2018 data. Remaining Land refers to $CO_2$ emissions from the LULUCF sector belonging to all six management classes[6] (Forest land, Cropland, Grassland, Wetlands, Settlements and Other Land) that remained for the previous 20 years. Data is found in the CRF Table 4,*

*points 4A1, 4B1, 4C1, 4D1, 4E1 and 4F1, (e.g. Austria CRF https://unfccc.int/documents/65597). The Converted Land refers to $CO_2$ emissions from conversions to and from all six classes that occurred in the previous 20 years, as reported in the CRF Table 4, points 4A2, 4B2, 4C2, 4D2, 4E2 and 4F2. HWPs are Harvested Wood Products and are reported in CRFs Table 4, point 4G. Bioenergy emissions are reported as a memo item under the Energy sector (CRF Table 1s2). These emissions are reported as decrease in carbon stock change in the LULUCF sector, and thus by convention*

*not accounted in the Energy sector.*

        Emissions from LULUCF represented in 2016 a sink of about 300 Mt $CO_2$, and this sink has increased 15 % from 1990 to 2016. Bioenergy emissions are reported as a memo in the energy sector, as the emissions are captured already under LULUCF.

For $CH_4$, the two largest anthropogenic sources in EU28 are agriculture (e.g. emissions from enteric fermentation) and waste (e.g. anaerobic waste) sectors. These two sources accounted for 90 % of total EU28 $CH_4$ emissions in 2016 excl. LULUCF (EEA, 2018) with Agriculture accounting for 53% of total EU28 $CH_4$ emissions in 2016 excl. LULUCF, that is 11 % of total EU28 GHG emissions excl. LULUCF in 2016. We exclude $CH_4$ emissions from LULUCF because they only represent 1.5 % of total EU28 $CH_4$ emissions in 2016. From 1990 to 2016, the total

$CH_4$ emissions from EU28 decreased by 31 % (554 Mt $CO_2$-eq). The top five EU28 emitters of $CH_4$ are France (13 %), Germany (12 %), UK (12 %), Poland (11 %) and Italy (10 %) that account for 56 % of total EU28 $CH_4$ emissions (excl. LULUCF sector).

---

[6] The IPCC Good Practice Guidance (GPG) for Land Use, Land Use Change and Forestry (IPCC 2003) describes a uniform structure for reporting emissions and removals of greenhouse gases. This format for reporting can be seen as "land based"; all land in the country must be identified as having remained in one of six classes since a previous survey, or as having changed to a different (identified) class in that period.

For N$_2$O, the largest EU28 sources are agriculture and the industrial processes and product use (IPPU) sectors, while the FOLU sub-sectors that cover emissions from forests are a small N$_2$O source. Agriculture contributes emissions largely from the use of fertilizers in agricultural soils, while industrial production of nitric and adipic acid dominates IPPU-related emissions. These sources accounted for 85 % of N$_2$O emissions in 2016, that is 5 % of total EU28 GHG emissions estimates in 2016. From 1990-2016, the total N$_2$O emissions decreased by 35 % (251 Mt CO$_2$-eq). The top five EU28 emitters of N$_2$O are France (18 %), Germany (16 %), UK (9 %), Poland (8 %) and Italy (8 %) that account for 59 % of the total N$_2$O EU28 emissions (excl. LULUCF sector).

Zooming on trends, non-CO$_2$ emissions show a very small decrease (- 0.4 %) from 2004 to 2014 and an increase (+0.8 %) from 2015 to 2017 (Olivier and Peters, 2018). This recent growth is principally determined by the increase of N$_2$O emissions which have offset the declining CH$_4$ emissions. The continued CH$_4$ emissions decrease is mainly due to shifts in the fossil fuel production from coal to natural gas in Germany, Italy and the Netherlands (BP, 2018).

The main objective of the present study is to present a synthesis of AFOLU GHG emission estimates from bottom-up approaches that can serve as a benchmark for future assessments, important during the reconciliation process with top-down GHG emission estimates. We use existing officially reported data from NGHGI submitted under the UNFCCC as well as other emission estimates based on research data, from global emissions datasets to detailed biogeochemical models. The bottom-up approaches considered, although based on independent efforts form those in the NGHGI, have some level of redundancy among them and the inventories, since they often use similar activity data (AD) and largely apply the current IPCC (2006) methodology, albeit using different 'Tiers'.

Independent bottom-up estimates are valuable to compare with estimates officially reported to the UNFCCC and may identify differences that need closer investigation. The uncertainties presented in this paper are taken from the UNFCCC NGHGI 2018 submissions. For the global emissions dataset EDGAR uncertainties are only calculated for the year 2012 as described in the Appendix B. We evaluate the reason for differences in emissions by carefully comparing the estimates, quantifying uncertainties and detecting discrepancies. We compare the inconsistencies (defined by differences between estimates) to the uncertainties (error associated to each estimate) and identify those sectors that would yield most benefit from improvements. Uncertainties from the other datasets and models were not yet available. We do include natural CH$_4$ emissions from wetlands, whose accounting will become mandatory from 2026 under the new EU LULUCF Regulation[7].

**2. Compilation of AFOLU emission estimates**

We collected available data of AFOLU emissions and removals (Table 1) between 1990 and 2016 (or last available year) that have been documented in peer reviewed literature. The collection of data represents the latest data

---

[7]https://eur-lex.europa.eu/legal-content/EN/TXT/?uri=uriserv:OJ.L_.2018.156.01.0001.01.ENG&toc=OJ:L:2018:156:FULL).

available and most recent state of the art of available estimates of GHGs representing the AFOLU sector in Europe as derived from our knowledge of the scientific literature and the scientific networks in Europe. UNFCCC NGHGI and other data sources for AFOLU emissions or component fluxes as well as methodologies are described in Appendix B. For all three GHGs, total emissions from Agriculture and LULUCF for EU28 are presented in Appendix Table A2.

Whenever necessary we provide details on individual countries separating $CO_2$, $CH_4$ and $N_2O$. The units are based on the metric ton (t) [$1kt = 10^9$ g; $1Mt = 10^{12}$g] for individual gases and [$Mt = 10^{12}$ g; $1Tg=10^{12}$g] for $CO_2$ and carbon (C) from AFOLU sectors. We rely on modelled and reported data-streams to quantify GHG fluxes from bottom-up models together with country specific inventory from NGHGI official statistics (UNFCCC), global inventory datasets (EDGAR), global statistics (FAOSTAT) and global land GHG biogeochemical models used for

research assessments (e.g. DGVMs, bookkeeping models). The values in this study are defined from an atmospheric perspective, which means that positive values represent a source to the atmosphere and negative ones a removal from the atmosphere.

*Table 1: Summary of AFOLU data sources for the three main GHG available and their references. In bold is*

*highlighted the last reported year for each underlying database used in this study.*

| *Official and other estimates* **(global datasets, models used for research)** | | | | | | |
|---|---|---|---|---|---|---|
| **CO₂** | | | | | | |
| *Data sources* | UNFCCC NGHGI 2018 (1990-**2016**) | CBM (2000-**2015**) | EFISCEN (1995-**2015**) | FAOSTAT (1990-**2016**) | Eight DGVMs TRENDY.v6 (1990-**2016**) | Bookkeeping model H&N (1990-**2015**) | Bookkeeping model BLUE (1990-**2017**) |
| *References* | 2006 IPCC GLs and CRFs | Pilli et al., 2016b, 2017 | Petz et al., 2016 | Tubiello et al. (2013) FAO, 2015 Federici et al., 2015 Tubiello, 2019 | Global Carbon Budget (GCB) 2017 (Le Quéré et al., 2018) | Houghton and Nassikas (2017) | Hansis et al., 2015 as updated for Le Quéré et al., 2018 |
| **CH₄** | | | | | | |
| *Data sources* | UNFCCC NGHGI 2018 (1990-**2016**) | EDGAR v4.3.2 (1990-**2012**) | EDGAR FT2017 (1990-**2016**) | CAPRI v. Star 2.3 (1990-**2013**) | FAOSTAT (1990-**2016**) | GAINS scenario "ECLIPSE v6" (1990-**2015**) | Natural (wetlands) CH₄ emission model ensemble Global Carbon Project (GCP) 2018 (1990-**2017**) |

| References | 2006 IPCC GLs and CRFs | Janssens-Maenhout et al., 2019 Crippa et al., 2019 | Olivier and Peters, 2018 | Britz and Witzke, 2014 Weiss and Leip, 2012 | Tubiello et al. 2013 FAO, 2015 Tubiello, 2019 | Höglund-Isaksson, L. 2012 Höglund-Isaksson, L. 2017 Gomez-Sanabria, A. et al., 2018 Höglund-Isaksson, L. 2020 | Poulter et al. 2017 GCP, 2018 |
|---|---|---|---|---|---|---|---|
| | | | | $N_2O$ | | | |
| Data sources | UNFCCC NGHGI 2018 (1990-**2016**) | EDGAR v4.3.2 (1990-**2012**) | EDGAR FT2017 (1990-**2016**) | CAPRI v. Star 2.3 (1990-**2013**) | FAOSTAT (1990-**2016**) | GAINS (1990-**2015**) | |
| References | 2006 IPCC GLs and CRFs | Janssens-Maenhout et al., 2019 Crippa et al., 2019 | Olivier and Peters, 2018 | Britz and Witzke, 2014 Weiss and Leip, 2012 | Tubiello et al. 2013 FAO, 2015 Tubiello, 2019 | Winiwarter et al., 2018 | |

As an overview of potential uncertainty sources, Appendix Tables A1a and A1b present the use of emission factor data (EF), activity data (AD) and, whenever available, uncertainty estimation methods used for all agriculture and forestry data sources used in this study. The referenced data used for the figures' replicability purposes are available for download at https://zenodo.org/record/3662371#.Xkui-WhKjIU (Petrescu et al., 2020). The complete emissions data can be found and downloaded from the source websites, as described in Appendix A, Tables A1a and A1b.

**3. Emission estimates**

As part of the AFOLU sectors, agricultural activities play a significant role in non-$CO_2$ GHG emissions (IPCC SRCCL 2019; FAO 2015). The two major gases emitted by the agricultural sector are $CH_4$ and $N_2O$. According to the 2018 UNFCCC NGHGI data updated up to the year 2016, agriculture contributes as much as 11 % from the total EU28 GHG emissions expressed in $CO_2$ equivalents (year 2016, UNFCCC NGHGI 2018). In 2016, $CH_4$ from agricultural activities accounted for 53 % of total EU28 $CH_4$ emissions, while $N_2O$ accounted for 78 % of $N_2O$ emissions respectively. The preponderant share of agriculture in total anthropogenic non-$CO_2$ emissions also applies

globally (IPCC SRCCL 2019). The $CO_2$ emissions reported as part of the agriculture sector only cover the liming and urea application, IPCC sectors 3G and 3H respectively. In terms of $CO_2$ they only represent <5 % of the total GHG emissions from agriculture, therefore are not included in this study.

Regarding the forestry sub-sector of AFOLU, LULUCF, the major GHG gas is $CO_2$. According to NGHGI 2018 data, in 2016, the total EU28 LULUCF sector was a net $CO_2$ sink of 314 Mt $CO_2$. We note that in general the reported values for GHG emissions do not include the flux estimates from LULUCF which are usually accounted for separately, because they are inherently very uncertain and show large inter-annual variations as a result of inter-annual variability in climatic conditions, and (in part as a consequence of this variability) in natural disturbances (Kurz et al., 2010, Olivier et al., 2017).

### 3.1. Agriculture $CH_4$ and $N_2O$ emissions

At EU28 level, GHG emission reporting is mandatory for all countries and is done under the consistent framework of UNFCCC. Every year in May all EU parties report to the convention their National Inventory Report (NIR) and provide data using the standardised Common Reporting Format (CRF) tables. The NIRs contain detailed descriptive and numerical information on all emission sources and the CRF tables contain all GHG emissions and removals, implied EFs and AD for the whole time series 1990 to two years before the submission year (https://unfccc.int/process-and-meetings/transparency-and-reporting/reporting-and-review-under-the-convention/greenhouse-gas-inventories-annex-i-parties/national-inventory-submissions-2018). It is important to note that the 2006 IPCC GLs used for this process do not provide methodologies for the calculation of $CH_4$ emissions and $CH_4$ and $N_2O$ removals from agricultural soils and field burning of agricultural residues. Parties that have estimated such emissions should provide, in the NIR, additional information (AD and EF) used to derive these estimates and include a reference to the section of the NIR in the documentation box of the corresponding sectoral background data tables.

Further in this section, we present estimates of $CH_4$ and $N_2O$ agriculture fluxes during the period from 1990 up to the last available year reported by each of the data sources. The detailed values for the last available year are shown in Appendix A, Table A2.

*$CH_4$ emissions*

According to UNFCCC NGHGI data, in 2016 agricultural activities accounted for 53 % of the total $CH_4$ emissions in EU28. At EU28 level (Fig. 2), we found that the total agriculture $CH_4$ emissions are consistent in trends and values among sources. For the agriculture sector totals our results show a relatively good match between UNFCCC and the four other data sources, with the lowest estimate (CAPRI) within 15 % of the UNFCCC value. The differences pertain mostly to Tier use (e.g. CAPRI) and expert judgment on the choice of EFs (e.g. EDGARv4.3.2.). Considering that the 2016 UNFCCC total agriculture reported uncertainty is 10 %, we acknowledge this relative difference of up to 15 % to be important in the emissions reconciliation process. In Table 2 we present the allocation of emissions by sub-sector following the IPCC 2006 classification. Key categories, investigated in this study for $CH_4$ on the EU28

level, are CH$_4$ emissions from enteric fermentation, CH$_4$ emissions from manure management, rice cultivation and

agricultural residues.

*Table 2: Agricultural CH$_4$ emissions - allocation of emissions in different sectors by different data sources used in this study.*

| Data source/sectors | UNFCCC NGHGI 2018 | EDGAR v4.3.2 | CAPRI | GAINS* | FAOSTAT |
|---|---|---|---|---|---|
| Agriculture | 3.A Enteric Fermentation | 4.A. Enteric Fermentation | CH$_4$ENT Enteric Fermentation | Enteric fermentation and manure management* (meat and dairy cattle, sheep, pigs, poultry) | Enteric Fermentation |
| | 3.B Manure Management | 4.B Manure Management | CH$_4$MAN Manure Management | | Manure Management |
| | 3.C Rice Cultivation | 4.C Rice Cultivation | CH$_4$RIC Rice Cultivation | Rice Cultivation (RICE) | Rice Cultivation |
| | 3.F Field Burning of Agricultural Residues | 4.F Agricultural waste burning | n/a | Agricultural waste burning (WASTE_AGR) | Burning – Crop Residues |

*\*GAINS does not separate between CH$_4$ emissions from enteric fermentation and manure management.*

As a consequence of the similar trends and distribution of emissions to sectors presented in Table 2, we notice a small but consistent variability of total emissions between the five data sources (Fig. 2).

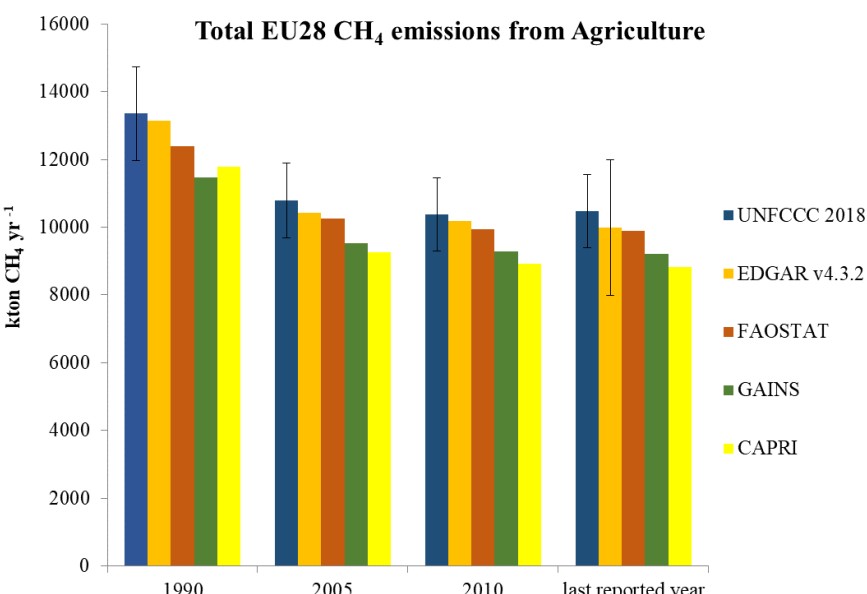

*Figure 2. Total EU28 Agriculture CH$_4$ emissions from five data sources, UNFCCC NGHGI 2018 submissions, EDGAR, FAOSTAT, CAPRI and GAINS. The relative error on the UNFCCC value, computed with the 95% confidence interval method, is 10 %. It represents the NGHGI 2018 uncertainty for the agriculture data reported to UNFCCC. Uncertainty for EDGAR v4.3.2 was calculated for 2012 and is 20 %; it represents the 95 % confidence interval of a lognormal distribution. Last reported year in this study refers to 2016 (UNFCCC and FAOSTAT), 2012 (EDGAR),*
*2015 (GAINS) and 2013 (CAPRI). The positive values represent a source.*

One possible cause for the similarity lays in the fact that almost all sources use EFs from the same IPCC GLs (2006). In EU28, AD are produced by four main sources and further dissiminated to the end users (see Fig. 4) and this can be subject to a certain amount of commonalities. Therefore, excluding AD and EFs, we might conclude that

differences shown in Figure 2 are mainly due to the choice of the Tier method for calculating emissions (e.g. in CAPRI as shown in Appendix A, Table A1a).

To better understand the differences between emissions in EU28 we plotted in Figure 3 the CH$_4$ emission percent difference between 2005 and 1990, and between the last reported year, 2010, 2012 (as the last common year reported by all sources) and 2005. We obeserve that for the 2005-1990 change there is a major reduction in CH$_4$

emissions for all data sources due to the implementation in the 1990s of European and country specific emission reduction policies on agriculture and environment, and socio-economic changes in the sector resulting overall into lower agricultural livestock, lower emissions from managed waste disposal on land and from agricultural soils. For the other three periods considered, the relative agricultural CH$_4$ reduction is smaller but still consistent between all data sources.

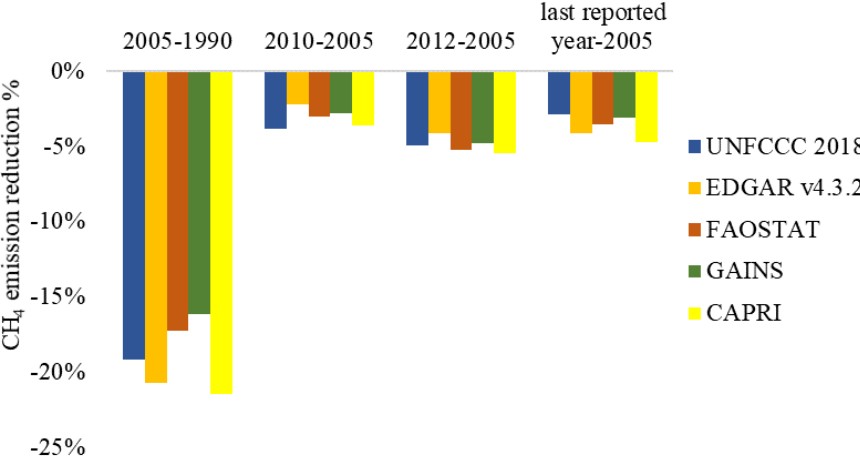

Figure 3: Change of EU28 total agricultural CH$_4$ emissions between different years. 2012 is the last common year when all sources have estimates. Last reported year in this study refers to 2016 (UNFCCC and FAOSTAT), 2012 (EDGAR), 2015 (GAINS) and 2013 (CAPRI).

We therefore conclude that all inventory-based data sources are consistent with each other for capturing recent CH$_4$ emissions reductions, or that they are not independent because they use similar methodology with different versions of the same AD (Fig. 4) which is mostly the case for the EU28 countries. The AD follows also a different course than the emissions data (see Fig. 4). The AD used is highly uncertain due to the collection process from surveys and different national reporting systems. FAOSTAT statistics use a relative value of 20 % uncertainty that is within the range for the confidence interval that IPCC (2006) suggests.

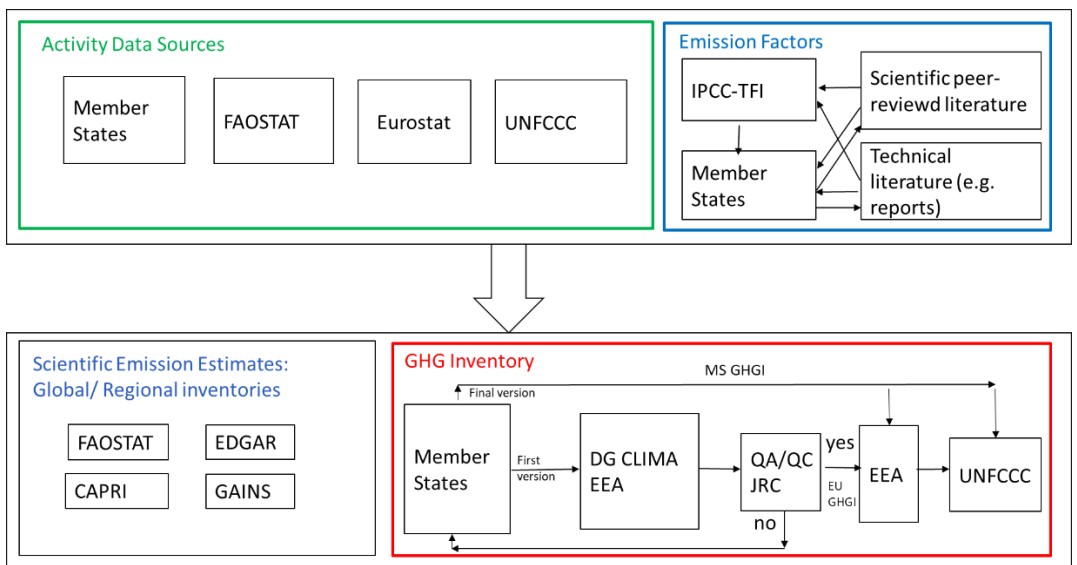

Figure 4: Example of flow of AD, EFs and emission estimates in EU based on IPCC regulations.

From the detailed analysis of CH4 emissions split into sectoral information (Figure 5) (all country data and figures are provided in the excel spreadsheets "Figures5,8_AppendixD_CH4_N2O_per_country" downloadable at https://zenodo.org/record/3662371#.Xkui-WhKjIU for the former Eastern European communist centralized economy block (Latvia, Lithuania, Estonia (ex USSR), Czech Republic, Poland, Romania and Hungary, East Germany) we notice very high CH4 emissions for 1990 which afterwards show a constant decreasing trend. This is best explained by the dissolution of the Soviet Union (1989–1991) and the consequent structural changes in their economy. The worst match between data sources in EU28 is found for Malta, Cyprus and Croatia but their emissions represent in the UNFCCC reporting less than 1 % of the total EU28 agricultural CH4 emissions. UNFCCC uncertainties for CH4 emissions are between 10-50 % but can be larger for some countries and sectors, e.g. Romania reporting a 500 % uncertainty for emissions from rice cultivation.

To exemplify the shares of CH4 emission from agriculture, in Figure 5 we present the total sub-sectoral CH4 emissions for three example countries.

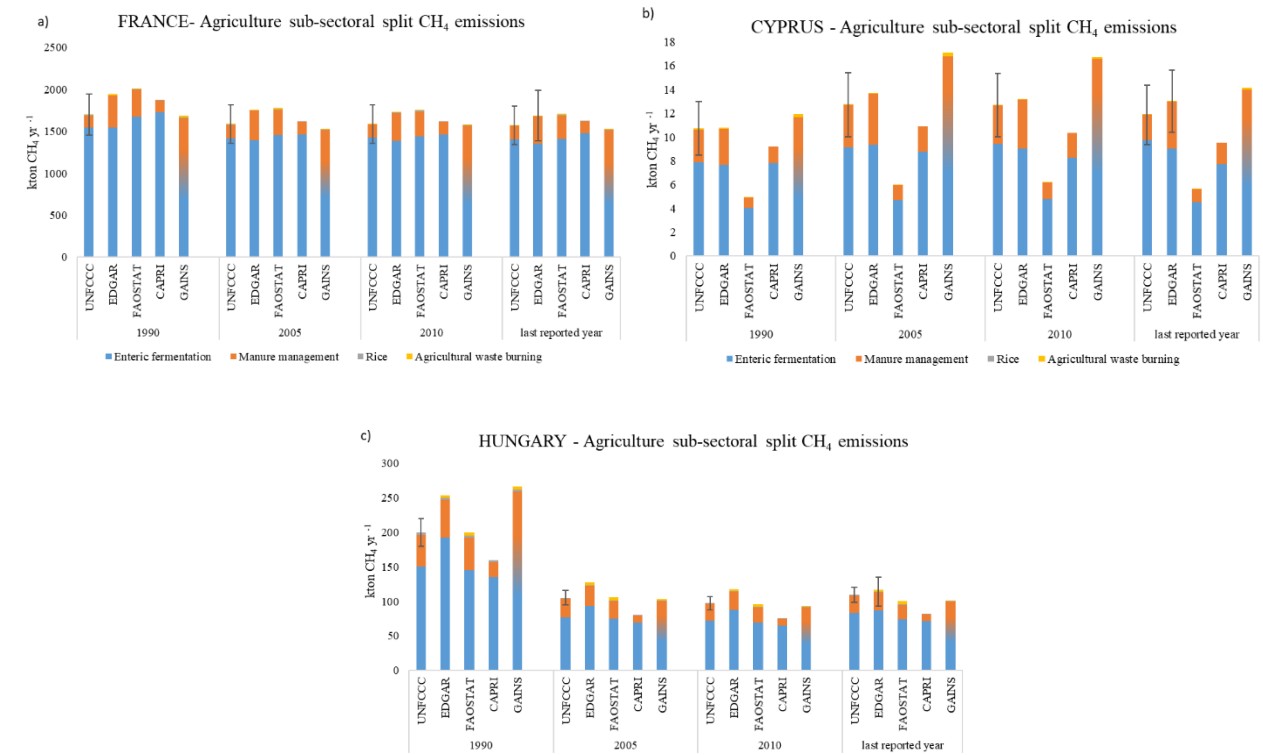

*Figure 5: CH4 emission from five data sources (UNFCCC NGHGI 2018, EDGAR v4.3.2., FAOSTAT, CAPRI and GAINS) split into main activities: enteric fermentation for ruminant livestock (blue) and manure management (orange). GAINS gradient (orange-blue) represent the total emissions from enteric fermentation and manure management. Rice cultivation and agricultural field burning banned since 2000 are very small and hardly distinguishable in the plots;* a) *very good consistency of the different data sources for France b) poor consistency for Cyprus; c) high 1990 CH4 emissions for Hungary (former Eastern European Block). The relative error on the*

*UNFCCC values are computed with the method described in Appendix C based on the NGHGI 2018 uncertainties for the agriculture CH₄ data reported to UNFCCC. Uncertainty for EDGAR v4.3.2 was calculated for 2012 and*
300 *represents the 95 % confidence interval of a lognormal distribution as discribed in Appendix B. The positive values represent a source. Last reported year in this study refers to 2016 (UNFCCC and FAOSTAT), 2012 (EDGAR), 2015 (GAINS) and 2013 (CAPRI).*

The highest share is attributed to enteric fermentation which for almost all countries count as ~80 % of total
305 agricultural CH₄ emissions. We notice that a very good consistency between emission estimates are found in Figure 5a for France while, on contrary a worse consistency is presented in Figure 5b for Cyprus, which might not report AD to FAOSTAT from its entire territory. Figure 5c exemplifies the high 1990 CH₄ emissions for Hungary in the former Eastern European Block and the lower subsequent estimates, mainly caused by political and economic changes after the dissolution of the Soviet Union (1989–1991). Note that some Eastern European countries, i.e. Romania and
310 Bulgaria, used different base years for Kyoto (1989 and 1988 respectively), as statistical data were considered problematic for 1990.

### *N₂O emissions*

According to UNFCCC NGHGI data, in 2016 agricultural activities accounted for 78 % of the total N₂O
emissions in EU28. For the agriculture sector, key categories on the EU28 level are N₂O emissions from manure management, direct N₂O emissions from agricultural soils and indirect N₂O emissions from agricultural soils. In Table 3 we present the allocation of emissions by subsector following the IPCC classification and we notice that each data source has its own particular way of grouping emissions.

*Table 3: Agricultural N₂O emissions - Allocation of emissions in different sectors by different data sources*

| Emission sources/Data providers | UNFCCC NGHGI 2018 | EDGAR v4.3.2 | CAPRI | GAINS | FAOSTAT |
|---|---|---|---|---|---|
| **Direct N₂O emissions from manure management** | 3.B.2 minus 3.B.2.5 – manure management | 4B – manure management | N₂OMAN – manure management | 3B – manure management | 3.B.2 – farming (N₂O and NMVOC emissions) |
| **Direct N₂O emissions** | 3.D.1.1 and 3.D.1.2 – direct N₂O emissions from managed soils | 4.D.1 – direct soil emissions | N₂OAPP – manure application on soils N₂OSYN – synthetic | 3.D.a.1 - Soil: Inorganic fertilizer and crop residues 3.D.a.2 - organic fertilizer | 3.D.1.1 - Inorganic N Fertilizers 3.D.1.2 - Organic N Fertilizers 3.D.1.4 – crop residues 3.D.1.6 – cultivation of organic soils |

| | | | | | |
|---|---|---|---|---|---|
| | 3.D.1.4 Crop residues 3.D.1.6 Cultivation of organic soils | | fertilizer application $N_2OHIS$ - histosols $N_2OCRO$ – crop residues | 3.D.a.6 - histosols | |
| **Direct and indirect $N_2O$ emissions from grazing animals** | 3.D.1.3 – Urine and Dung Deposited by Grazing Animals | 4.D.2 - Manure in pasture/range/paddock | $N_2OGRA$ - grazing | 3.D.a.3 - grazing | 3.D.1.3 -Urine and Dung Deposited by Grazing Animals |
| **Indirect $N_2O$ emissions** | 3.B.2.5. – Indirect $N_2O$ Emissions from manure management 3.D.2 Indirect emissions from soils | 4.D.3 – Indirect $N_2O$ from agriculture | $N_2OLEA$ - leaching $N_2OAMM$ – ammonia volatilization $N_2ODEP$ – atmospheric deposition (no IPCC) | 3.D.b.1 - atmospheric deposition 3.D.b.2 - leaching | 3.B.2.5 Indirect $N_2O$ emissions 3.D.2 - Indirect $N_2O$ Emissions From Managed Soils (atmospheric deposition and N leaching to the soils) |
| **Field burning of agricultural residues** | 3F - Field Burning of Agricultural Residues | 4F – agricultural waste burning | n/a | n/a | 3F - Field Burning of Agricultural Residues |

Similar to $CH_4$ emissions, $N_2O$ emissions show very good consistency between the five data sources for total EU28 emissions (Fig. 6). We note as well that uncertainties of UNFCCC and EDGAR are large but have similar magnitudes. Similar to $CH_4$, CAPRI has the lowest estimate but well within the uncertainty interval.

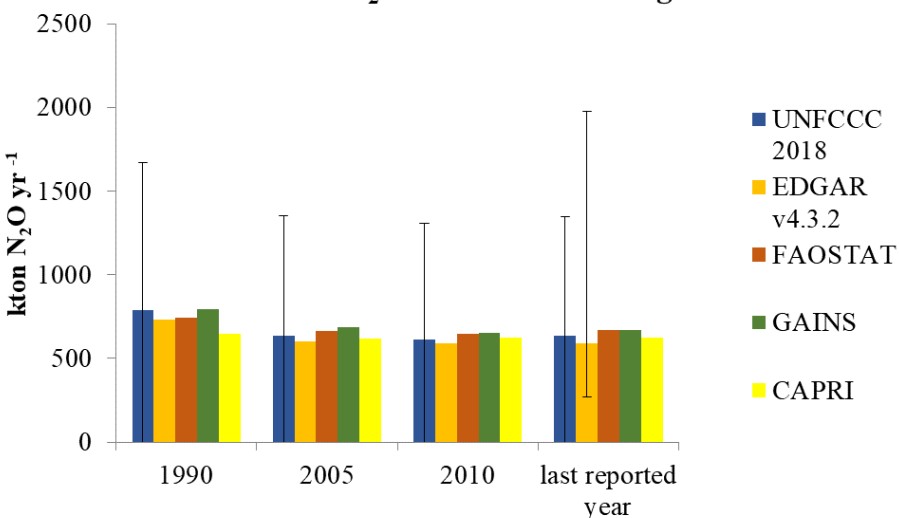

*Figure 6: Total EU28 Agriculture $N_2O$ emissions from five data sources, UNFCCC NGHGI 2018, EDGAR v4.3.2., FAOSTAT, CAPRI and GAINS. The relative error on the UNFCCC value, computed with the 95% confidence interval method, is 106 %. It represents the NGHGI 2018 uncertainty for the EU28 total $N_2O$ agriculture data reported to UNFCCC.. EDGAR uncertainty is only calculated for the last available year, 2012. Last reported year in this study refers to 2016 (UNFCCC and FAOSTAT), 2012 (EDGAR), 2015 (GAINS) and 2013 (CAPRI). The positive values represent a source.*

In Figure 7 we present the $N_2O$ emission difference between 2005 and 1990, and between the last reported year, 2012 (the last common year in reporting for all data sources), 2010 and 2005. We obeserve that for the 2005-1990 there is a major reduction in $N_2O$ emissions for all data sources for the same reasons stated for $CH_4$ but the spread between different reduction estimates is much larger then for $CH_4$. We do not see the same agreement for the reduction between 2010, 2012 and 2005 (i.e. CAPRI shows a small increase and other datasets a net decrease) and between last reported year and 2005 (i.e. FAOSTAT and CAPRI show small increases). The differences between the last reported year and 2005 could be partly attributed to the fact that the data sources have a different last reported year (see Table 1, in bold)

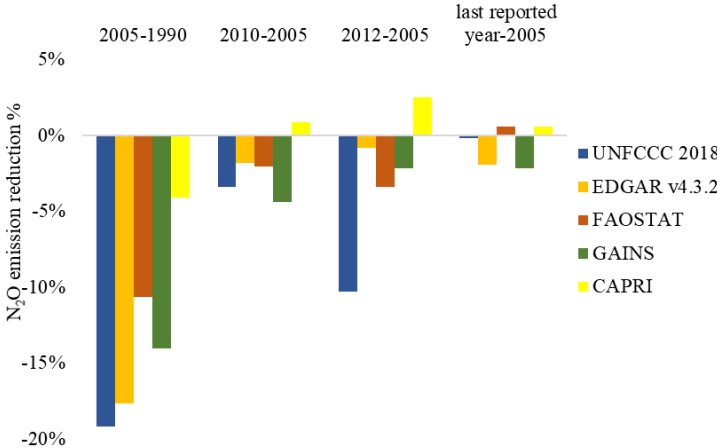

Total EU28 N$_2$O emissions reduction from Agriculture

*Figure 7: Change of EU28 total agricultural N$_2$O emissions between different years. 2012 is the last common year when all sources report estimates. Last reported year in this study refers to 2016 (UNFCCC and FAOSTAT), 2012 (EDGAR), 2015 (GAINS) and 2013 (CAPRI).*

Nevertheless, despite inconsistent sign of N$_2$O emission changes between datasets, the spread between absolute values of N$_2$O emission changes is smaller for recent periods than for the period 1990-2005. For both CAPRI and FAOSTAT, the increase in N$_2$O emissions, well represented by the positive changes seen in Figure 7, can be explained by changes in AD from synthetic fertilizers and correlated increment of crop residues.

The two most important sources for N$_2$O emissions from agriculture pertain to direct (synthetic fertilizer, manure application to soils, histosols, crop residues and biological nitrogen fixation) and indirect (ammonia vollatilization, leaching and atmopsheric deposition) emissions. We exemplify this in Figure 8 where we present the N$_2$O split in sub-activities.

We notice for the Eastern European former communist centralized economy block ((all country data and
figures are provided in the excel spreadsheets "Figures5,8_AppendixD_CH$_4$_N$_2$O_per_country.xlsx" downloadable at https://zenodo.org/record/3662371#.Xkui-WhKjIU (e.g. former USSR countries i.e. Latvia, Lithuania, Estonia and former Easter European Block i.e. Romania, Hungary, Slovakia, Bulgaria) higher N$_2$O emissions for 1990 which afterwards show a constant decreasing trend. This is again best explained by the economic transition in 1989–1991 and consequent impacts on the agriculture sector. The poorest consistency between data sources in EU28 is seen for
Belgium, Estonia, Lithuania, Latvia and Luxembourg ("Figures5,8_AppendixD_CH$_4$_N$_2$O_per_country.xlsx") but their emissions count for as much as 4.5 % of total EU28 N$_2$O emissions. In general, the uncertainties reported to UNFCCC for total N$_2$O emissions from the agriculture sector are very high and have a range between 22 % (Malta) and 207 % (Romania). For sub-activities extreme uncertainties are reported by Denmark and Bulgaria as 300 % for N$_2$O emissions from manure management, while Greece reports a very small uncertainty of less than 2 % for N$_2$O
emissions from

agricultural soils.

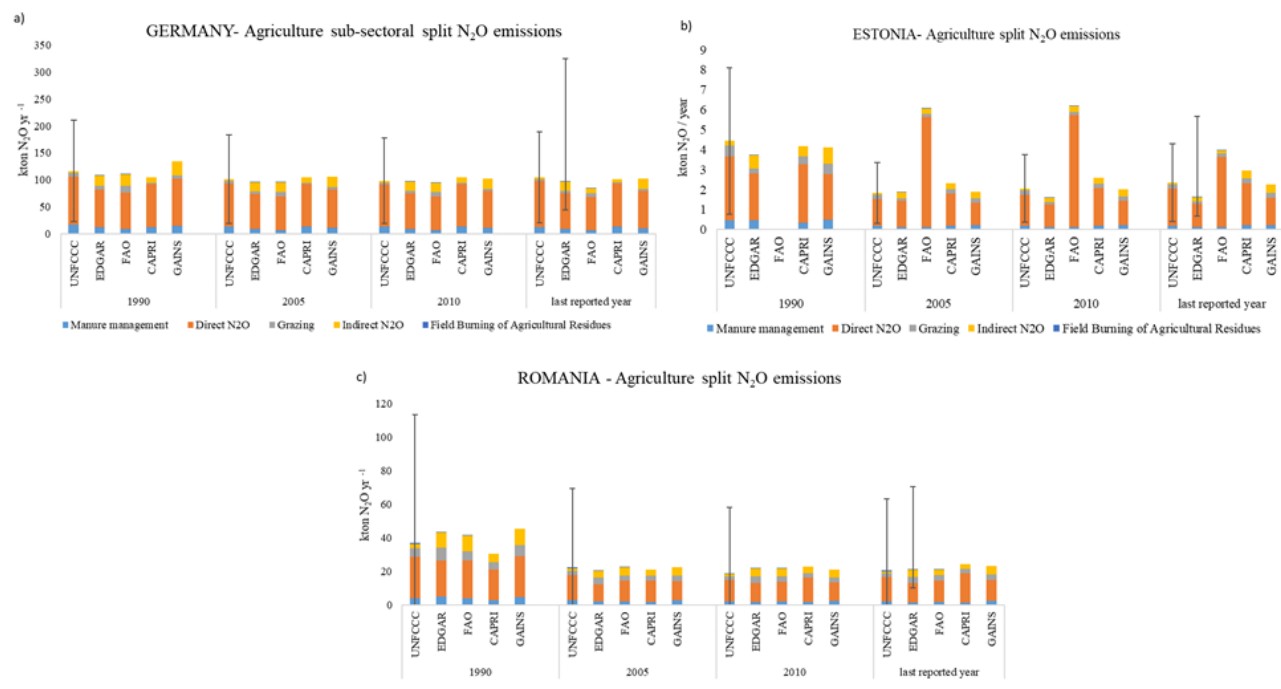

*Figure 8: N₂O emission from Agriculture split into main activities: manure management, direct emissions, grazing, indirect emissions and field burning of agricultural residues;* a) *very good consistency for Germany b) poor consistency for Estonia; c) high 1990 N₂O emissions for Romania (former Eastern European Block). The relative error on the UNFCCC values are computed with the method described in Appendix C based on the NGHGI 2018 uncertainties for the agriculture N₂O data reported to UNFCCC. Uncertainty for EDGAR v4.3.2 was calculated for 2012 and represents the 95 % confidence interval of a lognormal distribution as discribed in Appendix B. The positive values represent a source. Last reported year in this study refers to 2016 (UNFCCC and FAOSTAT), 2012 (EDGAR), 2015 (GAINS) and 2013 (CAPRI).*

EDGAR is using data from FAOSTAT thus, for the majority of countries (figures found at DOI: link as described in Appendix D), we observe similar estimates between these two sources (e.g. France, Italy, Poland). A reason for discrepancies may be attributed to the different way the data sources allocate their emissions to sub-activities (Table 3). For example, CAPRI N₂OSYN – synthetic fertilizer application - does not have a correspondent in GAINS activities. The leaching, ammonia and atmospheric deposition N₂O emissions in CAPRI do not have a clear correspondent sub-activity in UNFCCC, while in FAOSTAT those N₂O emissions are reported under other categories: manure left on pasture and manure applied to soils.

For N₂O emissions, uncertainties are mostly in the range of 100 % or more. The countries reporting the highest N₂O uncertainties are Buglaria, Denmark, Estonia and Cyprus, which, for manure management and agricultural soils, count as much as 200 % to 300 %. We notice that a very good match between emission estimates is found in Figure 8a for Germany while on contrary a worse match is presented in Figure 8b for Estonia with no FAO

data available in 1990 (only for former USSR). Figure 8c exemplifies the high 1990 $N_2O$ emissions for Romania (former Eastern European Block) and is due to irregularities in reporting during the dissolution of the Soviet Union (1989–1991).

### 3.2. Natural $CH_4$ emissions

In recent assessments of the global $CH_4$ budget (Saunois et al., 2019), wetlands $CH_4$ emissions from top-down and bottom-up estimates for the period 2008-2017 are statistically consistent and average 178 Tg $CH_4$ $yr^{-1}$ (range 155-200) and 149 Tg $CH_4$ $yr^{-1}$ (range 102-182), respectively (Saunois et al., 2019).

    In the EU28, natural emissions of $CH_4$ are represented by wetlands which are not yet fully accounted for and reported under NGHGI, their emissions reporting being only recommended under the 2013 IPCC Wetlands
Supplement (IPCC, 2014) complement to IPCC 2006 GL. According NGHGI 2019, between 2008 and 2017, the natural $CH_4$ emissions in the EU28 reported under LULUCF (CRF table 4(II) accessible for each EU28 country[8]) summed up to 0.1 Tg $CH_4$. The only countries in EU28 reporting $CH_4$ from wetlands were Denmark, Finland, Germany, Ireland, Latvia and Sweden.

    Wetlands are sinks for $CO_2$ and sources of $CH_4$. Their net GHG emissions therefore depend on the relative
sign and magnitude of the land–atmosphere exchange of these two major GHGs. Undisturbed wetlands are thought to have a large carbon sequestration potential because near water-logged conditions reduce or inhibit microbial respiration, but $CH_4$ production may partially or completely counteract carbon uptake (Petrescu et al., 2015). The net GHG balance of natural wetlands is thus uncertain. Natural emission of $CH_4$, in particular wetlands and inland waters and their net GHG balance, are the most important source of uncertainty in the methane budget (Saunois et al, 2019),
due to the GWP100 of $CH_4$ and the generally opposite directions of $CO_2$ and $CH_4$ fluxes.

    Under the new EU LULUCF Regulation article 7 (footnote 7), the accounting of natural wetland emissions will become mandatory from 2026 onwards, i.e. the reported numbers will be compared to numbers already reported under category 4(II) wetlands between 2005-2009 and the net difference will count towards reaching the EU climate targets.

Since $CH_4$ emissions are highly variable in time and space as a function of climate and disturbances, it makes EF-based methods impractical and national budget estimates difficult, making it challenging to accurately estimate $CH_4$ emissions in NGHGI. There is also a risk of double counting with emissions from inland waters as discussed e.g. by Saunois et al. 2019 for the global $CH_4$ budget. The sum of all natural sources of $CH_4$ as inferred by different models may be too large by about 30 % compared to the constraint provided global inversions. The spread of wetland
emissions from process-based wetland emission models used in the global $CH_4$ budget (Poulter et al., 2017) forced by the same variable flooded area dataset, is 30 % (80 Tg $CH_4$ $yr^{-1}$) globally (given their estimated emissions of 177–284 Tg $CH_4$ $yr^{-1}$ using bottom-up modeling approaches) and up to 70 % for EU28 calculated based on the model-to-model variability and even larger at national scale. In absence of any better information, we used in this study the results of these ensemble models (see Appendix B) to provide a first estimate of this source.

---

[8] https://unfccc.int/process-and-meetings/transparency-and-reporting/reporting-and-review-under-the-convention/greenhouse-gas-inventories-annex-i-parties/submissions/national-inventory-submissions-2018)

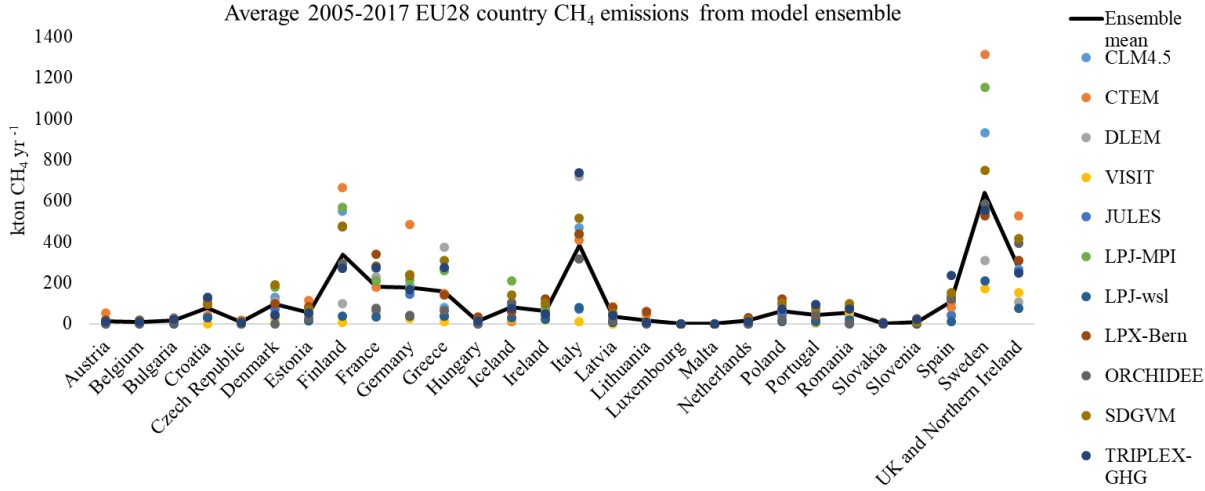


*Figure 9. Distribution of CH₄ emissions from undisturbed natural wetlands for all the countries of EU28 as simulated by an ensemble of 11 global emission models averaged between 2005-2017 (Poulter et al., 2017). The positive values represent a source. The models are explained in the acronym list and referenced in Appendix B.*

According to Poulter et al. (2017), between 2005-2017, the total wetland $CH_4$ emissions in EU28 averaged 3 Tg $CH_4$ with an uncertainty (1-sigma spread) of 70%, with seven countries having the highest emissions (Fig. 9). Finland, Italy, Sweden, UK, France, Greece and Germany, accounted for 75% of total EU28 wetland $CH_4$ emissions. For the same period, NGHGI 2019 reports an average of 10.34 kton $CH_4$ (0.01 Tg $CH_4$),a highly underestimated value compared to the modelled results, due to non-reporting and accounting under NGHGI.

Given this current gap between modelled and NGHGI reported data on $CH_4$ emission from wetlands in EU28, we stress the need of investing in better modelling methodologies for emission calculation and verification. Out of all EU28 countries, for the purpose of reporting, only Finland developed its own biogeochemical $CH_4$ model to provide to NGHGI a very detailed list of estimates for all $CH_4$ sub-activities.

**3.3. Forestry and Other Land uses**

          This Forestry and Other Land Uses, referred to here as LULUCF section, includes $CO_2$ emissions and removals from forests (incl. soils and harvested wood products) and soil organic carbon (SOC) changes from grasslands and croplands. A comprehensive assessment of the overall carbon stocks and fluxes of forests, would need

to be complemented by the analysis of climate change impacts on forest productivity and composition (Lindner et al., 2015). Several studies analysed the European forest carbon budget from different perspectives and over several time periods using GHG budgets from fluxes, inventories and inversions (Lyussaert et al,, 2012), flux towers (Valentini et al., 2000), forest inventories (Liski et al., 2000, Nabuurs et al., 2018, Pilli et al., 2017) and IPCC GLs (Federici et al., 2015).

Achieving the well-below 2ºC temperature goal of the PA requires, among other, negative emission technologies, low-carbon energy technologies and forest-based mitigation approaches (Grassi et al., 2018, Nabuurs et

al. 2017). Currently, the EU28 forests act as a sink  and forest management will continue to be the main driver affecting the productivity of European forests for the next decades (Koehl et al., 2010). Forest management, however, can enhance (Schlamadinger et al., 1996) or weaken (Searchinger et al., 2018) this sink. Furthermore, forest management not only influences the sink strength, it also changes forest composition and structure, which affects the exchange of energy with the atmosphere (Naudts et al., 2016), and therefore the potential of mitigating climate change (Luyssaert et al., 2018; Grassi et al., 2019).

We compared net $CO_2$ emissions and removals from the LULUCF sector reported by UNFCCC NGHGI 2018 to those included in FAOSTAT and to the carbon balance here termed as the Net Biome Production (NBP) from different models (Table 4). Categories presented in this study are forest land, cropland and grassland. We present separately the results from forest land and land use, because some models (e.g. CBM and EFISCEN) use a different definition of forest land than the DGVMs ensemble TRENDY (Sitch et al., 2008, Le Quéré et al., 2009) or bookkeeping models (Houghton & Nassikas 2017, Hansis et al., 2015).

To better illustrate differences between estimates we exemplify how four of the data sources interpret and calculate the NBP:

- UNFCCC NBP definition depends on the method used by each country;
- CBM calculates NBP as the total ecosystem stock change calculated as the difference between net ecosystem production (NEP) and the direct losses due to harvest and natural disturbances (e.g., fires) (Pilli et al., 2017, Kurz et al., 2009). Adding to the NBP the total changes in the harvested wood product (HWP) carbon stock, CBM estimates the net sector exchange (NSE) (Karjalainen et al., 2003, Pilli et al., 2017);
- EFISCEN's NBP is derived from total tree gross growth minus (density related) mortality minus harvest, minus turnover of leaves, branches and roots. From input of litter  minus decomposition, the soil balance is calculated with the Yasso soil model. Natural disturbances  tend to occur relatively little in Europe and, when happening, are included in regular harvest, therefore EFISCEN does not consider them in addition for the NBP calculation;
- DGVMs calculate NBP as the net flux between land and atmosphere defined as photosynthesis minus the sum of plant and soil heterotrophic respiration, carbon fluxes from fires, harvest, grazing, land use change and any other C flux in/out of the ecosystem (e.g. DIC, DOC, VOCs). Land use change emissions are calculated as the imbalance between photosynthesis and respiration over land areas that followed a transition. NBP should be equal to changes in total carbon reservoirs. The net land use change flux is derived by differencing the NBP of a simulation with and without land use change.

*Table 4: Model description and their references therein.*

| LULUCF data sources | Short description | References |
|---|---|---|
| UNFCCC CRF tables | Reported by Annex I (essentially developed) countries following the IPCC methodological guidelines (IPCC, 2006). | IPCC, 2006 |

| FAOSTAT | Tracks net carbon stock change in the living biomass pool (aboveground and belowground) associated with forests and net forest conversion to other land uses, using country specific emission factors (carbon densities) reported from countries to FAO following the IPCC stock difference method (IPCC, 2006) with FAOSTAT and FRA activity data from countries. It also contains estimates of $CO_2$ emissions from drained organic soils in cropland and grasslands; as well as non-$CO_2$ emissions from biomass fires other than agriculture and $CO_2$ and non-$CO_2$ emissions from fires on organic soils. | FAO, 2014 <br> Federici et al., 2015 <br> Tubiello, 2019 <br> Rossi et al. 2016 <br> Prosperi et al. (2020) in press (for fire emissions) <br> Tubiello et al. (2016) for peatland drainage |
|---|---|---|
| CBM | An inventory-based, yield-data driven model that simulates the stand- and landscape-level forest carbon dynamics of living biomass, dead organic matter and soil, including natural and anthropogenic disturbances. | Kurz et al., 2009 <br> Pilli et al., 2016 <br> Pilli et al., 2017 |
| EFISCEN | Empirical forest scenario simulator. It uses national forest inventory data as a main source of input. Includes a detailed dynamic growth module, while natural mortality and harvesting are included as regimes, depending on the region. | Verkerk et al., 2016 <br> Schelhaas et al., 2007 <br> Nabuurs et al., 2018 |
| BLUE | A half degree grid bookkeeping model that tracks individual histories of successive LULCC events in each grid cell. Estimates | Hansis et al., 2015 <br> Le Quéré et al., 2018 |

| | for peat burning and peat drainage are included. | |
|---|---|---|
| H&N | A country-level bookkeeping model, that tracks land use and land cover (croplands, pastures, plantations, industrial wood harvest, and fuelwood harvest) in four carbon pools (living aboveground and belowground biomass; dead biomass; harvested wood products; and soil organic carbon). Estimates for peat burning and peat drainage are included. | Houghton & Nassikas, 2017 |
| DGVMs (TRENDY v6) | Results of eight DGVMs presented in the GCB 2017 with variations in the land surface coverage of each model. Positive flux is into the land. | Le Quéré et al., 2018 |

*Forest Land*


Net $CO_2$ emissions/removals from Forest Land (FL) (in UNFCCC NGHGI 2018, IPCC sector 4A) includes net $CO_2$ emissions/removals from forest land remaining forest land and conversions to forests, i.e. it includes effects from both environmental changes and from land management and land use change as long as they occur on forest land declared as managed. According to IPCC GLs, to become accountable in the NGHGI under forest land remaining

forest land, a land must be a forest for at least 20 years. Over FL we compare modelled NBP estimates (presented as $CO_2$ net sink) simulated with CBM and EFISCEN models with UNFCCC and FAOSTAT data consisting of net carbon stock change in the living biomass pool (aboveground and belowground biomass) associated with forest and net forest conversion including deforestation.

Figure 10 presents the total net $CO_2$ sink estimates simulated with CBM and EFISCEN models (described in

Table 4 and Appendix B), FAOSTAT and countries official reporting done under UNFCCC. The sign convention is that negative numbers are a sink. The results show that the differences between models are systematic, with EFISCEN and CBM showing systematically lower sinks than UNFCCC, while FAOSTAT has systematically higher sinks and the FAOSTAT sink is increasing with time. The similarities between EFISCEN and CBM models are that they use national forest inventory (NFI) data as the main source of input to describe the current structure and composition of

European forest. However, CBM and EFISCEN make different assumptions about allometry, wood density or carbon content of trees. The difference between all estimates and FAOSTAT are probably because the stock change calculations directly the carbon stocks and area data computed by countries and submitted through the FAO Global

Forest Resource Assessments (FRA[9]), rather than employing models to estimate them. Further, FAOSTAT numbers include afforestation i.e. the sum of all other land converted to FL, while the others datasets do not, resulting in a smaller sink if afforestation is removed.

The UNFCCC NGHGI uncertainty of $CO_2$ estimates for FL at EU28 level, computed with the 95 % confidence interval method (IPCC, 2006) is 19.6 %, with uncertainty increasing to 25–50 % when analyzed at the country level (EU NIR 2014).Given that both CBM and EFISCEN use different methodologies to estimate emissions and removals (Pilli et al., 2016, Petz et al., 2016) likely leading to lower estimates than the NGHGI, we consider the match between the two models and the EU NGHGI to be satisfactory, given the uncertainties ad similarity in temporal trends.

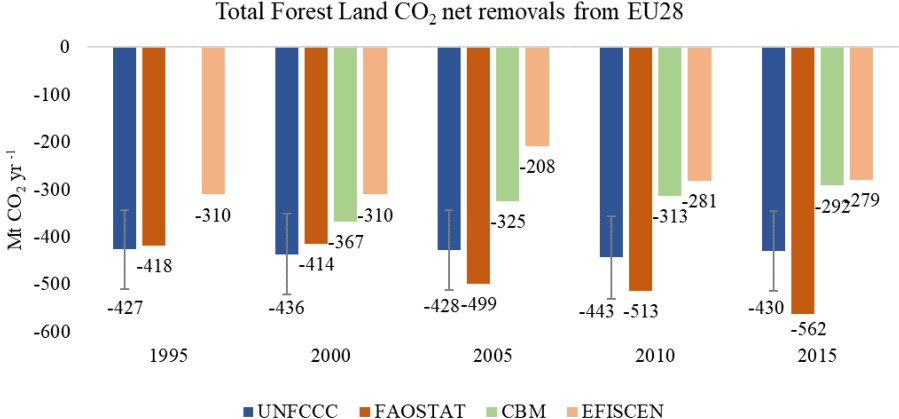

*Figure 10: Total EU28 single year values of $CO_2$ net removals from FL as reported by UNFCCC, CBM, EFISCEN and FAOSTAT. Negative numbers denote net $CO_2$ uptake. EFISCEN data 1995-2000 is based on Karjalainen et al., 2003 estimates. CBM does not report data for 1995. The relative error on the UNFCCC value, computed with the 95% confidence interval method, is 19.6 %. It represents the NGHGI 2018 uncertainty for the FL data pool reported to UNFCCC.*

From Figure 10 we see that while UNFCCC estimates are very stable, FAOSTAT shows an increasing sink, while CBM and EFISCEN show a saturating sink. And although all four are based on almost the same raw data, estimates differ by up to 50%. The sink of EFISCEN is somewhat lower because a higher harvesting was implemented in these runs. In 2015, most of the differences between FAOSTAT estimates and UNFCCC country data were generated by a few countries. For Finland, FAOSTAT reports around zero sink  and UNFCCC reports a large sink of 38 Mt $CO_2$ yr[-1]. For Romania and Latvia, the FAOSTAT sink is 165 Mt $CO_2$ yr[-1]and 17 Mt $CO_2$ yr[-1] respectively, a factor seven larger than the reported UNFCCC, 22 Mt $CO_2$ yr[-1] and 2.4 Mt $CO_2$ yr[-1] respectively. For Denmark, we find a sink according to FAOSTAT (-2.2 Mt $CO_2$) and a very small source reported to UNFCCC (0.17 Mt $CO_2$). When comparing NGHGI and FAOSTAT data, it should be considered that NGHGIs specifically report to the UNFCCC

---

[9]The Global Forest Resource Assessment (FRA) is the supplementary source of Forest land data disseminated in FAOSTAT, http://www.FAO.org/forestry/fra/en/

emissions and removals on managed forest land and are as such formally reviewed annually. By contrast, FAOSTAT emissions estimates include carbon stock changes over the total forest land area, and are not part of the UNFCCC formal reporting and review process (Grassi et al., 2017).

**Cropland and Grassland soil Carbon**

Cropland and Grassland (CL and GL) (in UNFCCC NGHGI 2018, IPCC sector 4B and 4C, respectively) include net $CO_2$ emissions/removals from soil organic carbon (SOC) under 'remaining' and 'conversion' categories. Similar as for FL, fluxes include effects from both environmental changes and from land management and land use change. FAOSTAT GHG emissions in the domain "Cropland" and "Grassland" are currently limited to the $CO_2$ emissions from cropland/grassland organic soils associated with carbon losses from drained histosols under cropland/grassland. This can be one of the reasons for differences between estimates reported by the two sources (Fig. 11).

Cropland definition in IPCC includes cropping systems, and agro-forestry systems where vegetation falls below the threshold used for the forest land category, consistent with the selection of national definitions (IPCC glossary). According to EUROSTAT, the term 'crop' within cropland covers a very broad range of cultivated plants. In 2015 more than one fifth (22 %) of the EU28's area was covered by cropland (EUROSTAT, updated in 2019). Denmark (51 %) and Hungary (44 %) had the highest proportion of their area covered by cropland in 2015. For the vast majority of the EU Member States (MS), cropland accounted for between 15 % and 35 % of the total area, this share falling to 10–15 % in Latvia, Estonia and Portugal, while the lowest proportions were registered in Slovenia (9 %), Finland (6 %), Ireland (6 %) and Sweden (4 %). In absolute terms, France, Germany, Spain and Poland had the biggest areas of cropland in 2015.

Grassland definition in IPCC includes rangelands and pasture land that is not considered as cropland, as well as systems with vegetation that fall below the threshold used in the forest land category.. This category also includes all grassland from wild lands to recreational areas as well as agricultural and silvo-pastural systems, subdivided into managed and unmanaged, consistent with national definitions. Grasslands tend to be concentrated in regions with less favorable conditions for growing crops or where forests have been cut down. Some of these are found in northern Europe (e.g. Finland and Sweden), while others are in the far south, i.e. the south of Spain.

In 2015 just above one fifth of the EU28's area (21 %) was covered by grassland. There is a broad range across EU Member States, with Ireland having 56% of its total land area as grassland and Finland and Sweden less than 6 % of the land (EUROSTAT, 2020).

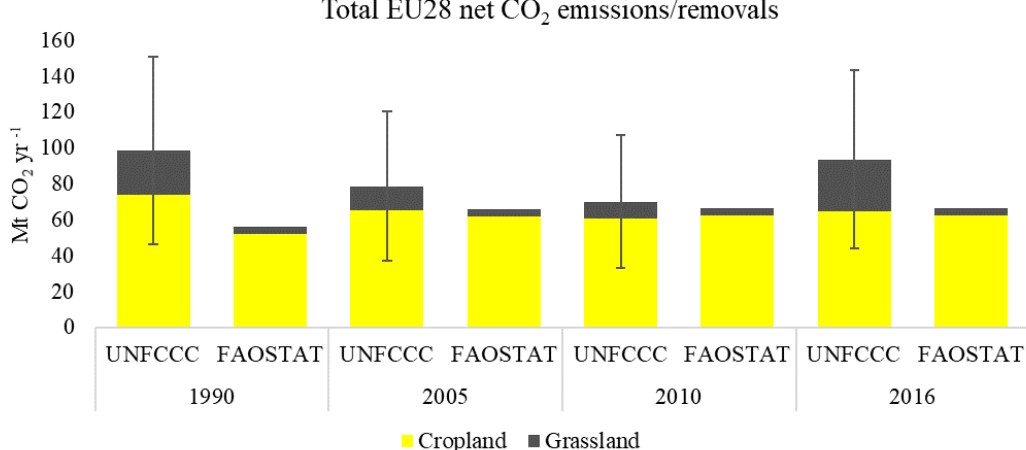

*Figure 11: Total EU28 net CO₂ emissions/removals from FAOSTAT and UNFCCC NGHGI 2018 submission estimates of Cropland and Grassland for 1990, 2015, 2010 and 2016. The relative error on the UNFCCC value, computed with the 95 % confidence interval method, is 53 %. It represents the NGHGI 2018 uncertainty for the CL and GL data pool*
*reported to UNFCCC.*

        Figure 11 shows that in the EU28 croplands and grasslands are $CO_2$ sources to the atmosphere in the UNFCCC NGHGI 2018 and FAOSTAT databases. Cropland $CO_2$ emissions are rather stable with time and are in good agreement between FAOSTAT and UNFCCC, except in 1990. Grassland emissions reported by countries to
UNFCCC are higher than the FAOSTAT and show an abrupt increase in 2016 compared to the previous years. The high estimates of grassland emissions in 2016 UNFCCC NGHGI submissions are explained by increased emissions in Austria, Denmark, Croatia; Sweden changed from being a sink in 2015 to being a very high source in 2016 and Hungary and Greece reported lower sink. Ireland was the only country which reported a higher sink in 2016 compared to 2015.
Climate change and climate effects on soil temperature and moisture are key drivers in the 21[st] century increase of soil decomposition and decrease of the soil carbon stock (Smith et al., 2005). Avoiding soil carbon losses or restoring stocks requires practices that increase C input in excess of losses from erosion and decomposition, such as diminished grazing intensity for grasslands, higher return of residues or reduced tillage for croplands, and manure additions for both. Further change in land use and management will also affect the soil carbon stock of European
cropland and grasslands (Smith et al., 2005).

**Land-related emissions from global models**

        Land-related carbon emissions can also be estimated by global models such as DGVMs (here we used the
TRENDY v6 ensemble) and two bookkeeping models (BLUE and H&N). In this section we compare these global model results with data from FAOSTAT and UNFCCC. There is significant uncertainty in both the underlying datasets of land-use changes, the coverage of different land use change practices, and the calculation of carbon fluxes . In

addition, marked differences in definitions must also be considered to compare independent estimates. Bookkeeping models give net emissions from land use change including immediate emissions during land conversion, legacy

emissions from slash and soil carbon after land use change, regrowth of secondary forest after abandonment, and emissions from harvested wood products when they decay. DGVMs estimate net land use emission as the difference between a run with and a run without land-use change, and their estimate includes the loss of additional sink capacity, that is, the sink that favors the environmental changes (e.g. $CO_2$ fertilization). This sink created over forest land in the simulation without land use change is "lost" in the simulation with land use change because agricultural land lacks

the woody material and thus has a higher carbon turnover (Gasser et al., 2013, Pongratz et al., 2014). This different definition from bookkeeping models historically implies higher carbon emissions from DGVMs, even if all post-conversion carbon stocks changes were the same in DGVMs and bookkeeping models.

The key difference between DGVMs and bookkeeping models, on the one hand, and FAO and UNFCCC methodology, on the other, is that the latter are based on the managed land proxy (Grassi et al., 2018a) (Fig. 12).


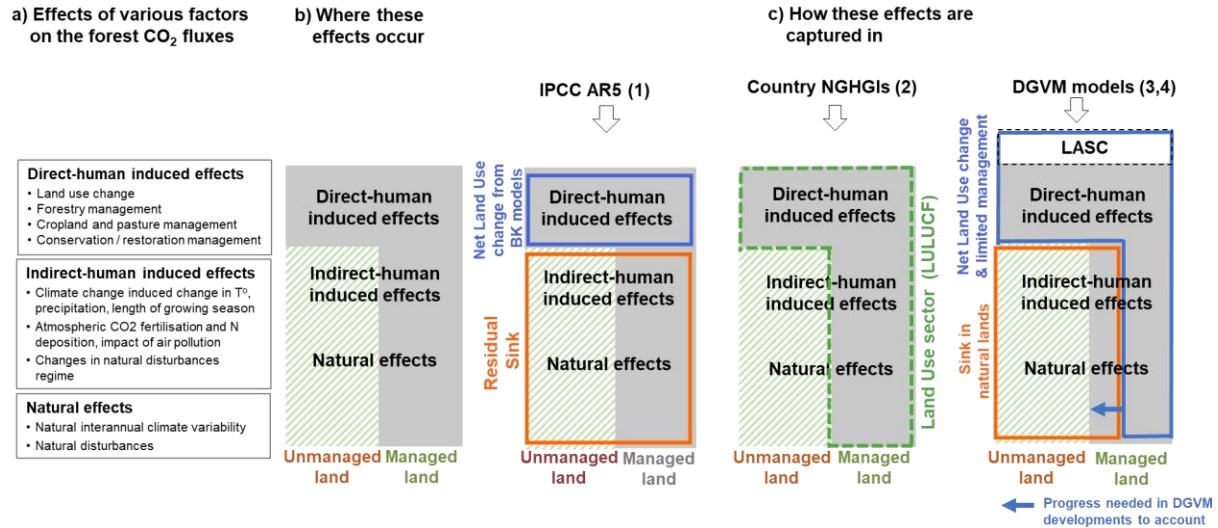

(1) In IPCC AR5, the residual sink is inferred as a difference between FF emissions + Net land use – growth rate – ocean uptake, and thus matches the observed $CO_2$ growth rate by construction. In this method, a bias on net land use change is transferred to the inferred residual sink
(2) In NGHGI, the LULUCF C balance only covers direct management actions and does not match the $CO_2$ growth rate. Any difference with the $CO_2$ growth rate can be attributed to errors in NGHGI estimates and / or fluxes on unmanaged lands
(3) In DGVM models, net land use change includes a source corresponding to the loss of additional sink capacity (LASC). Some models include limited land management (wood harvest, crop harvest). Non modelled management from forestry, cropland and pasture management, conservation / restoration management, being in the grey area part of the orange box
(4) DGVM models have parameterizations and structural uncertainties, and their net land flux does not match the global $CO_2$ growth rate, leading to a global BIM (budget imbalance)

*Figure 12: Summary of the main conceptual differences in defining the "anthropogenic land $CO_2$ flux" between IPCC 5th Assessment Report (AR5) and countries' GHG inventories (NGHGIs). a) Effects of key processes on the land flux as defined by IPCC; b) Where these effects occur (in unmanaged/primary lands vs. managed/secondary lands); c)*

*How these effects are captured: in the IPCC AR5 the anthropogenic "net land use" from Grassi et al., 2018a (solid blue line, including only direct human-induced effects), and the non-anthropogenic "residual sink" (solid red line, calculated by difference from the other terms in the GCB); countries' anthropogenic land flux from NGHGIs reported to UNFCCC (under the LULUCF sector, green dashed line), which in most cases includes direct and indirect human-induced and natural effects in an area of "managed" land that is broader than the one considered by Grassi et al.,*

*2018a. (Figure adapted from Fig. 3 in Grassi et al., 2018a); DGVMs modelled anthropogenic land flux (solid blue*

*line, including only direct human-induced effects and partly as managed land) and the non-anthropogenic "residual sink" (solid red line)partly covering the managed land. DGVMs simulate the net $CO_2$ flux from land use change by the difference between a simulation with variable land cover and a simulation with fixed land cover at the beginning of the simulation period. In the latter, ecosystems that are not converted are a foregone sink of $CO_2$ , causing the so-*

*called Loss of Additional Sink Capacity (LASC).*

Land fluxes can be differentiated into three processes (IPCC 2010): 1) Direct anthropogenic effects (land use and land use change, e.g., harvest, other management, deforestation), 2) Indirect anthropogenic effects (e.g., changes induced by human induced climate change, including $CO_2$ fertilization and nitrogen deposition changes), and 3) Natural effects (i.e., that would happen without human caused climate change, such as natural disturbances). ~~The~~

~~IPCC guidelines use a so-called land use proxy to estimate only "direct anthropogenic effects" that happen on managed land, hence how managed land is defined makes a big difference to reported emissions. In other words, following IPCC guidelines attributes all fluxes (including natural ones) on managed lands towards human activity.~~

~~In general, across all methods, managed land is defined as "land where human interventions and practices have been applied to perform production, ecological or social functions" (IPCC, 2006) but,~~ Models and GHGI capture

these effects in a different way:

**Biogeochemical Models**. *Bookkeeping approaches* only estimate direct anthropogenic effects. *DGVMs* also consider fluxes linked to indirect effects and natural processes. In the GCB 2018 (Le Quéré et al., 2018) and GCB 2019 (Friedlingstein et al., 2019), the fluxes associated to the direct anthropogenic effects are estimated with *Bookkeeping models and DGVMs,* while the remaining "land sink" (including all indirect and natural effects) are

estimated by *DGVMs*.

**National Greenhouse Gas Inventories (NGHGI)** use the notion of "managed land" as a proxy for "anthropogenic" emissions (IPCC, 2006), hence in practice include most or all (depending on the specific method) indirect emissions into their anthropogenic estimates. In addition, the area considered "managed" by countries is typically much greater than the area used by biophysical models to simulate the direct anthropogenic effects, as it

includes areas that are not actively managed (for instance, forest parks or forest seldomly harvested) (Grassi et al. 2018a).

The difference between biogeochemical models and NGHGI of around 4-5Gt $CO_2$ yr$^{-1}$ globally is to a large part attributable to the accounting of indirect effects on greater area managed land ~~towards AFOLU emissions for~~ by NGHGI compared to models (Grassi et al., 2018a, IPCC SRCCL 2019). The differences at the EU28 level are smaller,

because most forest land is considered managed by both models and NGHGI.

Independent estimates of the land-related flux for the EU28 are presented in Figure 13. The data behind the three main estimates, bookkeeping models, NGHGI and FAOSTAT represent the total net land use emissions/removal from forests, , including conversions to and from one category to another. Next to them, we plotted each of the net land use change flux (in grey) (difference of simulation with and without land use change) from eight of the

TRENDYv6 DGVMs used in the GCB 2017 (Le Quéré et al., 2018) with their mean, as they mostly simulate the indirect and natural sink considered unmanaged. FAOSTAT includes emissions from peatland drainage and fires, and from biomass fires (not considered herein). It does not include however other carbon stock changes in cropland and

grassland. We additionally excluded from UNFCCC estimate the categories Wetlands remaining Wetlands and Settlements remaining Settlements, as well as biomass burning and drainage, and transitions between non-forest lands.

The UNFCCC NGHGI and H&N's estimates are similar because the managed areas for EU28 are similar in both estimates (Grassi et al., 2018a). Differences between the two bookkeeping models, BLUE and H&N, relate to the different input data applied by each of the models and differences in biome types. The input used by H&N is based directly on FAOSTAT agricultural and wood harvest data and FRA forest area changes, while BLUE uses LUH2 (Hurtt et al., 2011, 2018). LUH2 is based on HYDE3.2 (Klein Goldewijk et al., 2017a, b), which provides annual,

half-degree, fractional data on cropland and pasture based on FAOSTAT, but overlays subgrid-scale transitions between all land use types and wood harvesting. H&N allocates pasture expansion preferentially on natural grasslands, while all available vegetation types of a grid cell are assigned proportionally to agricultural expansion in BLUE. Carbon densities and regrowth and decay curves are structurally similar, but differ in detail.

        The EU28 has a very small area of unmanaged land and this denotes that most of the LULUCF emissions in

the EU28 are from direct effects in the forestry sector (including agricultural expansion/abandonment). According to FAOSTAT and UNFCCC NGHGI, the net forest conversion is relatively small in the EU so the simulations include mostly managed net area.

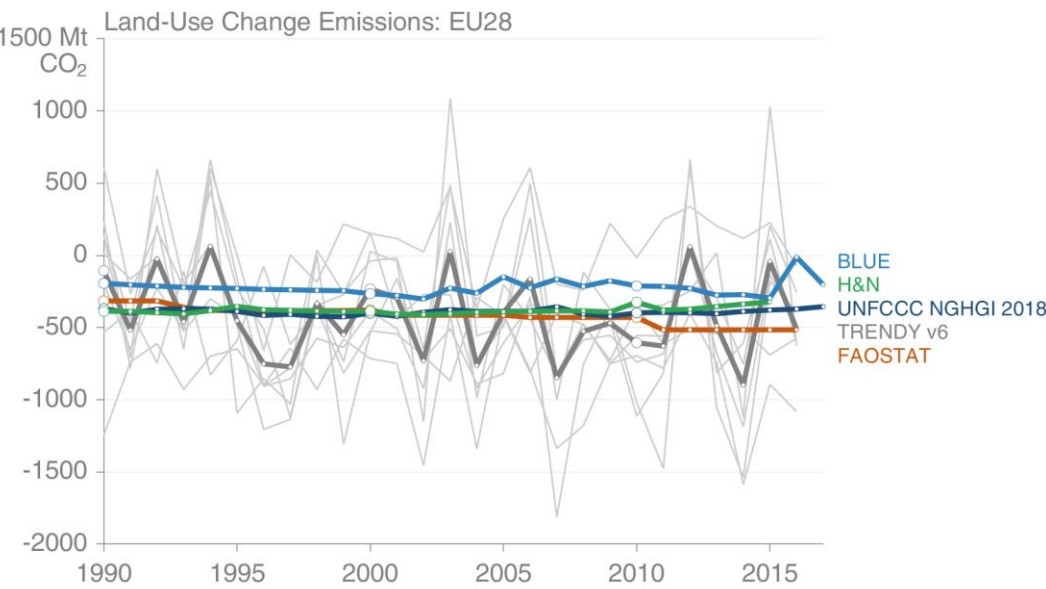

*Figure 13: A comparison of different estimates of the land use change flux in the EU28 from five data available*

*sources: BLUE, H&N, UNFCCC NGHGI 2018, DGVMs (TRENDY v6) and FAOSTAT. The grey lines represent the individual model data for eight DGVMs. The UNFCCC estimate includes the following categories: Forest Land, Cropland, Grassland net and with conversions and Wetlands, Settlements and Other land only conversions. The FAOSTAT estimate includes the following categories: Forest Land remaining Forest Land, afforestation and deforestation (conversion of forest land to other land types). The negative values represent a sink, while the positive*

*represent a source.*

DGVMs differ strongly in their estimate of the net land use change flux due to different comprehensiveness of including land use practices such as wood harvesting, shifting cultivation, or fire management (LeQuéré et al., 2018), different land use change datasets (HYDE3.2 or LUH2) and their implementation, on top of general model differences of how photosynthesis, respiration and natural disturbances are simulated. Most striking in comparison to the other, more empirical, approaches is the large inter-annual variability, related to the climate dependency of vegetation processes. Though DGVMs are conceptually similar to NGHGIs in simulating all indirect and direct fluxes on a given area, differencing of the simulations with and without land use change leaves only the land use related effects to be attributed to the net land use change flux (see Fig. 12). DGVMs are thus closer to the bookkeeping definition of LULUCF emissions, apart from differing assumptions on environmental changes (constant in bookkeeping, historical in TRENDY) and the loss of additional sink capacity included in DGVMs.

**4. Discussion**

4.1 Agricultural emissions

At European level the largest inconsistencies between estimates from AFOLU emission sources/sinks were found to be mainly caused by the use of different methodologies, including use of different AD and/or Tier level. When looking at final emission estimates, inconsistencies in methodology and Tier application in calculating emissions give as much as 10-20 % variation across estimates (e.g. $CH_4$ from agriculture),. Higher tiers require more detailed AD for calculating emissions/removals from AFOLU sectors.

Within the UNFCCC practice, for agriculture, each country uses its own country specific method which takes considers specific national circumstances (as long as they are in accordance with the 2006 IPCC GLs) as well as IPCC default values, which are usually more conservative. The EU GHG inventory underlies the assumption that the individual use of national country specific methods leads to more accurate GHG estimates than the implementation of a single EU wide approach (UNFCCC, 2018b). The Tier level a country applies depends on the national circumstances, which explains the variability of uncertainties among the sector itself as well as among EU countries. For example, inventory estimates of $N_2O$ emissions have very large uncertainties (>100 %) owing to the heterogeneity of sources and uncertainty in emission factors for the main $N_2O$ sources, in particular, agriculture. Since agricultural soil and manure management emissions vary strongly from site to site depending on e.g. soil properties and background emissions, management and meteorology, it is extremely challenging to determine accurate mean emission factors (JRC report, https://ec.europa.eu/jrc/en/publication/eur-scientific-and-technical-research-reports/atmospheric-monitoring-and-inverse-modelling-verification-greenhouse-gas-inventories). Winiwarter et al., 2018 stated that under current technologies, agricultural emissions have a large potential to abatement and, in the short term, reductions of $N_2O$ emissions must rely on the adoption of existing technologies. Currently available technology could reduce global $N_2O$ emissions by about 26 % below the baseline projection in 2030 (Winiwarter et al 2018). The most applicable pathways to enhance emission reductions are: the refinements of existing options (use of fertilizers), increasing the efficiency of measures (N use efficiency), changing human diets (lower consumption of animal protein). Oenema et al., (2013) estimate a total reduction potential for $N_2O$ emissions from agriculture including human diet changes of up

to 60 % in 2050, adding about half to the reductions available from technical measures alone (41 % reductions). According to Höglund-Isaksson et al. (2012) and the scenario work based on GAINS model, technical mitigation potential for the agriculture sector in 2030 can only reach 8% due to mitigation opportunities which are found limited and often costly both from social and private interest rates.

Concerning the IPCC calculation of $CH_4$ emissions from enteric fermentation, depending on the type of animal, the situation within the EU28 varies from country to country. For cattle (IPCC sector 3.A.1) emissions are calculated with very sophisticated methods, with only Cyprus using partially Tier 1. For the enteric fermentation of sheep (3.A.2), the situation is more diverse with 13 countries using Tier 1 methods and 15 using higher tiers (including those with higher emissions). For other cattle (3.A.4), only three countries (Romania, France and Portugal) are using higher tiers, with all the others combining different methods. $CH_4$ and $N_2O$ emissions from manure management (3.B.1 and 3.B.2) it is even more mixed, with Germany, Denmark, Finland, France, Croatia and Romania using exclusively higher tiers in both categories. For the calculation of emissions from soils, the share of high tiers is very low; only Denmark and Sweden use solely higher tiers in indirect $N_2O$ emissions from agricultural soils (3.D.2), while there are no countries using only high tiers in direct $N_2O$ emissions (3.D.1), but only some combining high with low tier methods (UNFCCC, 2018b). All these differences in calculating emissions produce evidently higher uncertainties in the results. For the UNFCCC, throughout the variability of the analyzed national GHG inventories, it turned out that $N_2O$ emissions from manure management and direct and indirect emissions together with $CH_4$ emissions from rice cultivation have the largest uncertainties. When we aggregated UNFCCC uncertainties at country level (using the methodology described in Appendix C), we also noticed the fact that not all countries report sub-sectoral uncertainties (e.g. Greece for grazing) and some countries (Sweden, Poland, Croatia and Czech Republic) had no uncertainty analysis performed for all sub-activities due to lack of data (e.g. confidential data).

There is as well the need to define a common methodology for overall uncertainty calculation while checking for consistency in the way uncertainties are calculated for different data sources and the way data is aggregated for different sectors. We noticed that for agricultural $N_2O$ emissions the split in sub-activities is not always consistent with IPCC sectors and this leaves room to differences when aggregating the results (Table 3).

### 4.2. Forestry and Other Land uses

For the LULUCF sector, methods for the estimation of GHGs and $CO_2$ fluxes still differ among countries and land use categories. Within the UNFCCC practice, strict Good Practice Guidance is prescribed, but there are still small differences between countries as each considers specific national circumstances (as long as they are in accordance with the 2006 IPCC GLs), as well as IPCC default values. When we analyze the estimates from multiple sources (inventories and models) we observe that, published estimates contain two main sources of uncertainties: a) differences due to input data and structural/parametric uncertainty of models (Houghton et al., 2012); b) differences in definition (Pongratz et al., 2014; Grassi et al., 2018b). These differences result from choices in the simulation setup, and are partly predetermined (for b) in particular) by the type of model used: bookkeeping models, DGVMs, or inventory-based – and whether fluxes are attributed to LULUCF emissions due to the cause or place of occurrence (indirect fluxes on managed land included in NGHGIs and FAOSTAT). Differences in definitions and methodology

calculation of estimates across model types is crucial and may lead to model-to-model variability. In Figure 13 the variability between the mean of the DGVMs ranges between 44 % in 1996 and 186 % in 2016 (distance between interquartile range and median across models for each year).

Depending on the degree of independence between assumptions, variability can become a reliable proxy for structural uncertainty when more accurate estimates are lacking (Solazzo et al., 2017). In general the definition of NBP denotes the net gain or loss of carbon from a region. NBP is equal to the Net Ecosystem Production (NEP) minus the carbon lost due to a disturbance (e.g. forest fire, harvest)) taking into account as well the net C balance of harvested products (described by the IPCC 2006 GLs) and C emitted by inland waters. In the context of land use change, the GCB 2017(Le Quéré et al., 2018) highlighted harvest as one of the main uncertainties. Only to exemplify, according Nabuurs et al. (2018) the uncertainty affecting all studies is that EU harvesting levels are rather uncertain. According to the FRA report 2015, most European countries have a solid forest inventory but there is still large uncertainty over harvesting levels. For many countries forest statistics from FAO have shortcomings such as: very large differences between reported periods, data corrected in later versions, unreported (harvest) removals (Nabuurs et al., 2018).

Checking collective progress towards meeting the goals of the PA will be done by the PA's global stocktake. At present, there is a discrepancy of about 4-5 Gt $CO_2$ $yr^{-1}$ in global anthropogenic net land use emissions (Grassi et al., 2018a, IPCCC SRCCL) between DGVM models reflected in IPCC assessment reports and aggregated national UNFCCC GHG inventories. Grassi et al., 2018a shows that about 3.2 Gt $CO_2$ $yr^{-1}$ can be explained by conceptual differences in anthropogenic forest sink estimation, related to the representation of environmental change impacts and the areas considered as managed. In order to limit the temperature increase to 1.5°C and keep it well below 2°C, as set by the PA, net-zero $CO_2$ emissions at global level need to be achieved around 2050 and neutrality for all other GHGs somewhat later in the century. At this point, any remaining GHG emissions in certain sectors need to be compensated for by absorption in other sectors, with a specific role for the land use sector, agriculture and forests (DG CLIMA Report, 2018).

It is important to distinguish between reporting and accounting in the GHG inventory context, as not all reported emissions account towards emission reduction efforts (Grassi et al., 2018b). Reporting refers to the inclusion of estimates of anthropogenic GHG fluxes in NIRs, following the methodological guidance provided by the IPCC. The NIR should, in principle, aim to reflect "what the atmosphere sees" (Peters et al., 2009) in managed lands, within the limits given by the method used and the data available. In the context of mitigation targets (e.g. the PA), accounting refers to the comparison of emissions and removals with the target and quantifies progress toward the target. For the LULUCF sector, specific accounting rules are used to filter reported flux estimates with the aim to better quantify the results of mitigation actions (Grassi et al., 2018b). The UNFCCC reporting principles allocate emissions to the territorial location (national boundaries) at the time that they occur (Peters et al., 2009).

The different definitions and concepts used by the global models and inventory communities mean that the land fluxes cannot necessarily be consistently compared. The framework developed by Grassi et al. (2018a) and shown in Figure 12 can be generalized to make a more direct comparison as applied to EU28 (Figure 14). Figure 14 disaggregates managed forest land into components that are reported in the UNFCCC CRFs: converted land (e.g., land changing from cropland to forest land), and the remaining land (e.g., forest land remaining forest land) is split into

land that is "production" (forestry $R_F$) or land that is used for "ecological or social functions" (other $R_o$), based on the definitions of managed land. Unmanaged land (sink, S) cannot have direct human induced effects.

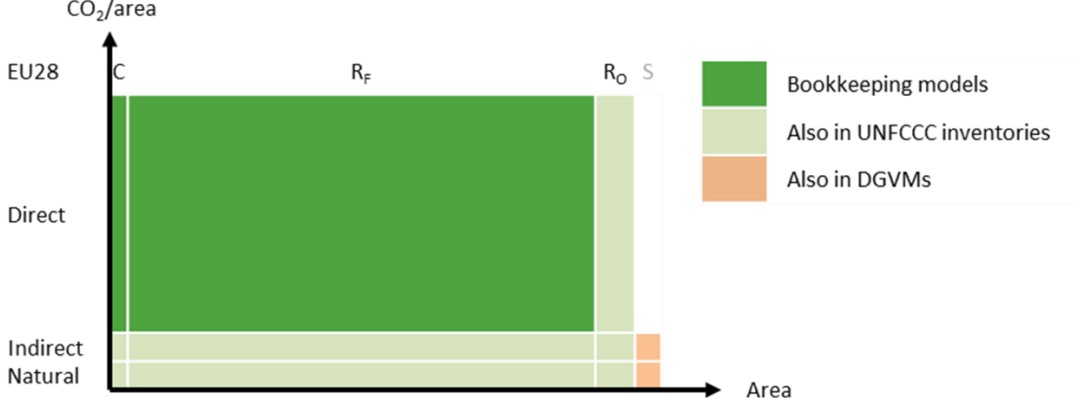

*Figure 14: A conceptual extension of Figure 12, applied to EU28, to disaggregate the managed land into the components reported in the UNFCCC inventories. The vertical axis represents density, horizontal axis area, and the area of each box is the $CO_2$ emissions. The diagram is conceptual and not to scale, but it does give an indication of how the components may look in the EU28. The converted land (C) is equivalent to afforestation plus deforestation.*

*Remaining land is split into remaining forestry (RF) and remaining other (ecological and social functions) (Ro) and the sink (S) belongs to unmanaged land. Bookkeeping models consider only direct effects (dark green), but do not include transitions between cropland and pasture or land management changes (increasing tillage, introducing irrigation etc.). UNFCCC CRFs include all dark and light green components (direct, indirect, natural on managed land) while DGVMs in principle can include all components including the sink (pink).*

Overall, our results suggest that most of the LULUCF emissions in the EU28 are from direct effects in the managed forest sector, including age-legacy effects (forest expansion and regrowth after WWII), with small net emissions from land conversion as they are largely compensated by deforestation (from CRFs). With appropriate data and models, it is theoretically possible to expand and enumerate the estimates more accurately.

**5. Conclusions**

There are many independent estimates of GHG emissions, but adequate understanding of their differences (either qualitatively or quantitatively) is lacking. For $CH_4$ and $N_2O$ emissions the main differences between country

reports and models are the use of tiers and methodologies (for both emissions and uncertainty calculation). Countries reporting to UNFCCC use an inconsistent mix of tiers depending on the animal type and activity following the approach described by the 2006 IPCC GLs, while models run with more accurate data and have better disaggregation of activities. One detected similarity between all sources is the use of EFs, as almost all sources make use of the IPCC defaults. AD is often shared, mostly sourced from the MS, FAOSTAT, Eurostat or UNFCCC, with the reasons for

differences in activity data between these four sources not totally understood.

At EU28 level, there is room to improve NGHGI's consistency between UNFCCC Tier use and models (e.g. for $CH_4$ from agriculture 10-20 % difference). We stress the need for more detailed quantification of difference between LULUCF $CO_2$ estimates (inventories, models etc.), caused by inconsistencies in methodology and/or Tier application. More data and analysis is needed to account for and reduce the differences in estimates. Narrowing down the analysis to sensitive parameters (e.g. AD) which may trigger the differences (e.g. Appendix A, Table A1) also requires more information on uncertainties.

It is of great importance to better distinguish between direct and indirect effects on land use emissions especially for the purpose of reconciling land-related emissions from global datasets and NGHGI. Currently our comparisons give significant uncertainty, mostly related to coverage of different land use practices and the differences in definitions (Fig. 12).

It is important to recognize that just because independent inventories agree well for a sector, does not necessarily mean that the estimate is closer to the actual emissions. The reason for agreement across inventories may simply be that the different inventories used the same methodology and data sources. In recent years there has been increased attention to the quantitative differences between land-based $CO_2$ emissions, with a much better understanding between inventories and estimates from the scientific community. However, there remain gaps in our understanding of differences between FAOSTAT and UNFCCC and between different DGVMs and bookkeeping models. One explanation can be linked to the fact that models use different methods to estimate emissions/removal then countries use in reporting to UNFCCC.

The current atmospheric GHG network is coordinated by the Integrated Carbon Observation System (ICOS) infrastructure at the European level. Within the future UNFCCC reporting framework, we argue that countries should use, whenever possible, global inversions to provide additional constraints for the verification and reconciliation purposes. A synthesis of available top-down non-$CO_2$ estimates has already been undertaken by Bergamaschi et al. (2015) and was not discussed here, but within the VERIFY project framework, we will use it in a following study focused on inversions based on better, higher resolution, transport models to assimilate the precise ICOS GHG concentration data complemented by satellite retrievals of column $CO_2$, $CH_4$ and $N_2O$ concentrations. While the GCP (Friedlingstein et al., 2019) provides the global carbon budget, this study starts a series of datasets for EU. These are essential for the GHG Monitoring and Verification Support (MVS) capacity, the EU envisages to build in support of the enhanced transparency framework of the Paris Agreement. The European Commission decided to take up a new service for monitoring anthropogenic $CO_2$ emissions under the long-term Copernicus, which is under construction (Janssens-Maenhout et al., 2020) and will make use of regional inversions, coupled with global inversions.

The main challenge for the inversion community remains the separation of natural and anthropogenic part of the total emission column. For the moment, global inverse models are widely used to estimate emissions of $CH_4$ and $N_2O$ at global/continental scale, using mainly high-accuracy surface measurements at remote stations (e.g. Bergamaschi et al., 2013; Bousquet et al.,2006; Mikaloff Fletcher et al., 2004a, b; Saunois et al., 2016, Hirsch et al., 2006; Huang et al., 2008; Saikawa et al., 2014; Thompson et al., 2014b; Wells et al., 2018, JRC report, 2018) with few regional inversions used to mainly estimate the European $CH_4$ and $N_2O$ emissions (Bergamaschi et al., 2015, 2018).

## 6. Appendices

### Appendix A

### Methodology tables

*Table A1a: Agriculture source specific activity data (AD), emission factors (EF) and uncertainty methodology*

| CH₄/N₂O emission calculation | AD/Tier | EFs/Tier | Uncertainty assessment method | Emission Data availability |
|---|---|---|---|---|
| UNFCCC | Country-specific information consistent with the IPCC GLs | IPCC GLs / Country specific information for higher Tiers | IPCC GLs (https://www.ipcc-nggip.iges.or.jp/public/2006 gl/) for calculating the uncertainty of emissions based on the uncertainty of AD and EF, two different approaches: 1. Error propagation, 2. Monte Carlo Simulation | NGHGI official data (CRFs) is found at: https://unfccc.int/process-and-meetings/transparency-and-reporting/reporting-and-review-under-the-convention/greenhouse-gas-inventories-annex-i-parties/submissions/national-inventory-submissions-2018 |
| EDGAR | IEA, FAOSTAT, USGS, WSA, IFA, NBS of China<br><br>Tier 2 (but when info is failing, Tier 1) | Mainly derived from IPCC defaults (Tier1). Depending upon availability of more refined estimates, country specific EF are adopted (Tier 2 and Tier 3) | IPCC GLs for emission factor and activity data uncertainty; assumptions for the propagation of the uncertainty when aggregating emission from several sources and/or countries. | Total and sub-sectoral EDGAR v4.3.2 GHG emissions are found at: https://edgar.jrc.ec.europa.eu/overview.php?v=432_GHG |
| CAPRI | Farm and market balances, economic parameters, crop areas, livestock population and yields from EUROSTAT, parameters for input-demand functions at regional level from FADN (EC), data on trade between world regions from FAOSTAT, policy variables from OECD. | IPCC 2006: Tier 2 for emissions from enteric fermentation of cattle and from manure management of cattle. Tier 1 for all other livestock types and emission categories. N-flows through agricultural systems (including N excretion) calculated endogenously. | n/a | Available at: https://zenodo.org/record/3662371#.Xkui-WhKjIU |
| GAINS | Livestock numbers by animal type (FAOSTAT, 2010; EUROSTAT, 2009; UNFCCC, 2010) Growth in livestock numbers from FAOSTAT (2003), CAPRI model (2009) Rice cultivation Land area for rice cultivation (FAOSTAT, 2010) Projections for EU are taken from the CAPRI Model | Country-specific information and: Livestock - Implied EFs reported to UNFCCC and IPCC Tier 1 (2006, Vol.4, Ch. 10) default factors Rice cultivation - IPCC Tier 1–2 (2006, Vol. 4, p. 5.49 Agricultural waste burning - IPCC Tier 1 (2006, Vol. 5, p. 5.20 | IPCC (2006, Vol.4, p.10.33) uncertainty range | Detailed country total CH₄ and N₂O emissions can be obtained by contacting the data providers: For CH₄: HÖGLUND ISAKSSON Lena hoglund@iiasa.ac.at For N₂O: WINIWARTER Wilfried winiwart@iiasa.ac.at |

| FAOSTAT | FAOSTAT Crop and Livestock Production domains; FAOSTAT Land use Domain; Harmonized world soil; ESA CCI; MODIS 6 Burned area products | IPCC GLs | IPCC (2006, Vol.4, p.10.33) Uncertainties in estimates of GHG emissions are due to uncertainties in emission factors and activity data. They may be related to, inter alia, natural variability, partitioning fractions, lack of spatial or temporal coverage, or spatial aggregation. | Agriculture total and sub-domain specific GHG emissions are found for download at: http://www.fao.org/faostat/en/#data/GT Land use total emissions and their subdomains for land use classes cropland, grassland, forest land and for biomass burning can be found at: http://www.fao.org/faostat/en/#data/GTL; |


*Table A1b: LULUCF source specific activity data (AD), emission factors (EF) and uncertainty information.*

| *CO₂/NBP emissions calculation* | *AD/Tier* | *EFs/Tier* | *Uncertainty assessment method* | *Emission data availability* |
|---|---|---|---|---|
| UNFCCC | Country-specific information consistent with the IPCC GLs | IPCC GLs / Country specific information for higher Tiers | IPCC GLs for calculating the uncertainty of emissions based on the uncertainty of AD and EF, two different approaches: 1. Error propagation, 2. Monte Carlo Simulation | NGHGI official data (CRFs) is found at: https://unfccc.int/process-and-meetings/transparency-and-reporting/reporting-and-review-under-the-convention/greenhouse-gas-inventories-annex-i-parties/submissions/national-inventory-submissions-2018 |
| CBM | national forest inventory data, Tier 2 | EFs directly calculated by model, based on specific parameters (i.e., turnover and decay rates) defined by the user | n/a used from IPCC | Available at: https://zenodo.org/record/3662371#.Xkui-WhKjIU and detailed emissions can be obtained by contacting the data providers: Giacomo.GRASSI@ec.europa.eu Roberto.PILLI@ec.europa.eu |

| EFISCEN | national forest inventory data, Tier 3 | emission factor is calculated from net balance of growth minus harvest | Sensitivity analysis on EFISCEN V3 in Schelhaas et al. 2007. (the manual) . Total sensitivity is caused by esp. young forest growth, width of volume classes, age of felling and few more. Scenario uncertainty comes on top of this when projecting in future. | Detailed country level emissions can be obtained by contacting the data providers: Gert-Jan Nabuurs gert-jan.nabuurs@wur.nl Mart-Jan Schelhaas martjan.schelhaas@wur.nl |
|---|---|---|---|---|
| FAOSTAT | The FAOSTAT emissions database is computed following Tier 1 IPCC 2006 GLs for National GHG Inventories (http://www.ipcc-nggip.iges.or.jp/public/2006gl/index.html). | The FAOSTAT emissions database is computed following Tier 1 IPCC 2006 GLs for National GHG Inventories (http://www.ipcc-nggip.iges.or.jp/public/2006gl/index.html). | n/a | FOLU total and sub-domain specific $CO_2$ emissions are found for download at: http://www.fao.org/faostat/en/#data/GL |
| DGVMs (TRENDYv6) | Can be considered as Tier 3 although the models have never been used for any reporting | Can be considered as Tier 3. Cover only LCC emissions for $CO_2$ | Model specific, SD of the annual $CO_2$ sink across the models as described in Le Quéré et al., 2018, section 2.6.2. | TRENDYv6 data is available for download at: http://dgvm.ceh.ac.uk/node/21/index.html Contact: Stephen Sitch S.A.Sitch@exeter.ac.uk |
| Bookkeeping models (H&N and BLUE) | Simple assumptions about C-stock densities (per biome or per biome/country) based on literature | Transient change in C-stocks following a given transition (time dependent EF after an land use transition) | There is no uncertainty estimate per model | H&N and BLUE country level data can be obtained by contacting the data providers: H&N: Richard Houghton rhoughton@whrc.org BLUE: Julia Pongratz julia.pongratz@geographie.uni-muenchen.de |

*Table A2: Total EU28 Agriculture and LULUCF estimates in kton gas per year reported by the five data sources for last available year (**in bold**).*

| EU28 | Year | Total EU28 LULUCF estimates for last available year kton $CO_2$, $yr^{-1}$ (Figures 10 and 13) | | | |
|---|---|---|---|---|---|
| | | Forest Land Remaining Forest Land | Cropland | Grassland | Total Land use |
| UNFCCC NGHGI 2018 | 2016 | -429000 | 64513 | 29101 | -371354 |
| FAOSTAT | 2016 | -562000 | 62291 | 4328 | -515350 |
| CBM | 2015 | -292000 | | | |
| EFISCEN | 2015 | -279000 | | | |
| TRENDY v6 mean | 2016 | | | | -511672 |

| | | | | | | |
|---|---|---|---|---|---|---|
| BLUE | 2017 | | | | -204104 | |
| H&N | 2015 | | | | -320649 | |

| EU28 | Year | Total EU28 Agriculture estimates for last available year kton CH₄ yr⁻¹ (Figures 2,3,5) | | | | |
|---|---|---|---|---|---|---|
| | | Enteric Fermentation | Manure management | Rice Cultivation | Agricultural Waste Burning | Total |
| UNFCCC NGHGI 2018 | 1990-**2016** | 8513 | 1832 | 105 | 25 | 10475 |
| EDGAR v4.3.2 | 1990-**2012** | 7576 | 2263 | 103 | 48 | 9990 |
| FAOSTAT | 1990-**2016** | 7630 | 1987 | 221 | 53 | 9893 |
| GAINS | 1990-**2013** | 9007 | | 97 | 105 | 9208 |
| CAPRI | 1990-**2015** | 7470 | 1269 | 86 | - | 8825 |

| EU28 | Year | Total EU28 Agriculture estimates for last available year kton N₂O yr- ((Figures 6,7,8) | | | | | |
|---|---|---|---|---|---|---|---|
| | | Manure management | Direct N₂O emissions | Grazing | Indirect N₂O emissions | Agricultural Waste Burning | Total |
| UNFCCC NGHGI 2018 | 1990-**2016** | 87 | 393 | 70 | 134 | 0.76 | 685 |
| EDGAR v4.3.2 | 1990-**2012** | 49 | 346 | 82 | 110 | 1.23 | 588 |
| FAOSTAT | 1990-**2016** | 73 | 381 | 94 | 117 | 1.4 | 667 |
| GAINS | 1990-**2013** | 67 | 392 | 76 | 135 | - | 670 |
| CAPRI | 1990-**2015** | 71 | 412 | 79 | 61 | - | 623 |

**Appendix B**
**Data source description**


**UNFCCC**

The UNFCCC committed in articles 4 and 12 in particular developed country parties listed in the Annex I of the UNFCCC to provide a national inventory of anthropogenic emissions by sources and removals by sinks of all greenhouse gases not controlled by the Montreal Protocol using comparable methodologies. The Conference of Parties
(COP) decided in 2013 (Decision 24/CP.19[10]) on revised UNFCCC Reporting Guidelines for Greenhouse Gas Inventories of Parties in Annex I to the Convention (UNFCCC 2013), here after UNFCCC Reporting Guidelines, based on the the 2006 IPCC GLs for National Greenhouse Gas Inventories (Eggleston, Buendia et al. 2006).

---

[10] https://unfccc.int/resource/docs/2013/cop19/eng/10a03.pdf#page=2

Developing countries are neither requested to mitigate GHG emissions nor to provide detailed information on national GHG emissions on an annual basis.

The exclusion of developing countries is explained by the fact that in the 1990s, when the Convention's and the Kyoto Protocol's reporting system was developed and adopted, a clear division of the regional distribution of GHG emissions existed. In industrialized countries, most GHG emissions were released, while in developing and emerging countries, emissions were low (Berger, Günther et al. 2016).

The UNFCC Reporting Guidelines decided to commonly use the Global Warming Potentials (GWP)[11] values
with a 100 year horizon of the IPCC 4[th] Assessment Report (IPCC 2007) for the calculation of emissions in order the compare differed greenhouse gases and to report the emission in complete time series from 1990 up to two years before the due date of the reporting, using the spreadsheets of the Common Reporting Format (CRF). The reporting is strictly source category based and divided into the following main sectors: Energy (CRF 1), Industrial processes and product use (CRF 2), Agriculture (CRF 3)[12], Land use, land use change and forestry (LULUCF) (CRF 4)[13,] and
Waste (CRF 5). For each sector, the CRF tables of the UNFCCC Reporting Guidelines provide a detailed catalog of source categories reflecting a comprehensive inventory of all sources and sinks of the above-mentioned gases within an economy. Together with the calculation and reporting of emissions by sources and removals by sinks, countries have to provide a mandatory assessment of the uncertainties of the data provided.

Chapter 3 of *2006 IPCC GLs for National Greenhouse Gas Inventories* on the mandatory Uncertainty Assessment provides two approaches for the uncertainty calculations in national greenhouse gas inventories:

1) error propagation
2) Monte Carlo simulations

For both approaches Chapter 3 use two main statistical concepts – the probability density function (PDF) and
confidence limits, where the probability density function describes the range and relative likelihood of possible values and the confidence limits give the range (confidence interval) within which the underlying value of an uncertain quantity is thought to lie for a specified probability.

Under Approach 1, there are two ways in which uncertainties can be calculated:

a) Where uncertain quantities are to be combined by multiplication, the standard deviation of the sum will be the
square root of the sum of the squares of the standard deviations of the quantities that are added.

b) Where uncertain quantities are to be combined by addition or subtraction, the standard deviation of the sum will be the square root of the sum of the squares of the standard deviations of the quantities that are added.

---

[11] The common metric Global Warming Potential (GWP) enables the comparison of different greenhouse gases by converting them into $CO_2$-equivalents.

[12] Whereas before 2015 no $CO_2$ emissions were reported under Agriculture, from 2015 the $CO_2$ emissions from urea and lime application were reallocated from LULUCF to Agriculture.

[13] The Revised UNFCC Reporting Guidelines keep within the IPCC AFOLU sector Agriculture and LULUCF distinguished. This represents a distinction between the UNFCCC Annex I reporting guidelines as determined in negotiation between parties and the UNFCCC, and the IPCC Reporting Guidelines.

For this study an analysis of the reported uncertainties under the NGHGI for $CO_2$, $CH_4$ and $N_2O$ has been performed for 26 EU countries[14]. The analysis has not been performed for Sweden and Czech Republic due to lack of data (e.g. confidential data). Due to lack of data availability only the Approach 1: propagation of error has been applied to each country's uncertainty assessment in order to identify the main uncertainties. The second approach (i.e. Monte Carlo simulations) was not used in this study, although presumably providing the more meaningful results.

Since the EU MS report all on different subsectors, the uncertainties have been aggregated to the subsectors per gas that all countries have in common, see the following table B1[15]:

*Table B1: Aggregation of sub-sectors for the uncertainty analysis.*

| | |
|---|---|
| Energy Sector (CRF 1A) | 1A1, 1A2, 1A3, 1A4, 1A5 |
| Fugitive Emissions Sector (CRF 1B) | 1B1, 1B2 |
| IPPU Sector (CRF 2) | 2A, 2B, 2C, 2D, 2E, 2F, 2G, 2H |
| Agriculture Sector (CRF 3) | 3A, 3B, 3C, 3D, 3E, 3F, 3G, 3H |
| LULUCF Sector (CRF 4) | 4A, 4B, 4C, 4D, 4E, 4F, 4G, 4H |
| Waste Sector (CRF 5) | 5A, 5B, 5C, 5D, 5E |

Generally, for almost all countries, the uncertainties for $CO_2$, $CH_4$ and $N_2O$ in the agriculture sector, LULUCF sector are rather high and variable compared to the other sectors. For the EU as a whole, the uncertainties vary by sector; for the agriculture sector it is 45.4%, and for the LULUCF sector it is 33% (UNFCCC 2018). This is because of the inherently different aspects of these sectors due to their dependencies on a number of variable factors and parameters, which make it harder to measure greenhouse gases accurately. For example, (Rypdal and Winiwarter 2001) claim that it is the incomplete understanding of soils that is the largest contribution to national uncertainty assessments, which can be confirmed with the uncertainty analysis[16].

**EDGAR**

The Emissions Database for Global Atmospheric Research (EDGAR) with versions EDGARv4.3.2. and EDGAR FT2017 provide global, country-level and gridded annual emissions of $CO_2$, $CH_4$ and $N_2O$ (as well as of other species, not discussed here), used by policy makers and the IPCC (AR5).

EDGAR is developed and maintained by the Joint Research Centre of the European Commission, with continued inputs by PBL. The version v4.3.2 released in 2017, (Janssens-Maenhout et al., 2019) provides 0.1° gridded emissions

---

[14] All MS analysed in this study have performed their uncertainty assessment using the approach 1, i.e. the methodology of propagation of error.

[15] All sectors and subsectors are covered, however the table explains which sub-sectors are aggregated for uncertainty calculation purposes.

[16] $N_2O$ emissions in soil are affected by microbiological activity and processes, the natural variation in soil conditions and the impacts of inter-annual variation in climate on the emissions, making it difficult to measure. Other important contributions to the overall uncertainty are uncertainties about the amount of solid waste (organic material that decomposes to produce methane) that is deposited and the extent of land use change.

from 1970 to 2012. The 'Fast Track' (FT) version produced every year using a variant method provides time series updates making use of latest available information on major sources (energy statistics of IEA and BP).

The EDGAR v4.3.2FT2015 has been producing 2015 grid maps at 0.1°x0.1° resolution for the H2020 project $CO_2$ Human Emissions (CHE). The agriculture component of EDGAR comprises the agricultural soils (crops that are not rice) ($N_2O$), application of urea and agricultural lime ($N_2O$), enteric fermentation ($CH_4$), rice cultivation ($CH_4$), manure management ($CH_4$, $N_2O$), fertilizer use (synthetic and manure) ($N_2O$), agricultural waste burning (in field) ($CH_4$, $N_2O$) and is based on agricultural statistics and commodity statistics for some products (e.g. lime). Although

agricultural field burning is included, other large-scale biomass burning from Savannah and forests and carbon stock changes due to land use activities are not included in EDGAR (Janssens-Maenhout et al., 2019). Details on EDGAR methodology for emissions calculations and uncertainties is referenced in Table A1.1a Recently, EDGAR v4.3.2FT2015 has been updated to EDGAR v5/v4.3.2FT2017 (Olivier and Peters, 2018) which includes national $CH_4$ and $N_2O$ emissions up to 2017.

EDGAR uses emission factors (EFs) and activity data (AD) to estimate emissions. Both EFs and AD are uncertain to some degree, and when combined their uncertainties need to be combined too. To estimate EDGAR's uncertainties (stemming from lack of knowledge of the true value of the EF and AD), the methodology devised by IPCC (2006, Chapter 3) is adopted, that is the sum of squares of the uncertainty of the EF and AD (uncertainty of the product of two variables). When aggregating the emissions from subcategories, or different sources, or countries the

covariance of the respective probability distribution enter into play.

The assumptions introduced by, e.g. Bond et al, (2004), Bergamaschi et al, (2015), Olivier et al., (2002) hold:

- Uncertainties of different source categories are uncorrelated;

- Subsectors for $CH_4$ and $N_2O$ are fully correlated, thus the uncertainty of the sum is the sum of the uncertainties;

- When dealing with $CO_2$, full correlation is assumed for subsets sharing the same emission factors (typically fuel-dependent);

- Aggregated emissions from same categories but different countries assumes full correlation, unless the emission factor is country-specific, or derived from higher tiers (i.e. not default EF defined by IPCC).

In addition, the following assumption is adopted:

- When uncertainty is defined within a range (e.g. for the energy sector, IPCC recommend that the methane emission factors are treated with an uncertainty ranging from 50% to 150%), the upper bound of the range is assigned to developing countries, whilst the lower bound to developed countries. Uncertainty of country or process-specific EF is not propagated (no correlation).

Although assuming full correlation when aggregating emissions is quite conservative (overestimating the

uncertainty introduced by emission factors), this approach is intended to balance for other sources of uncertainty that are not taken into account, such as covariance among activity data (deemed negligible), uncertainty of technologies factors (no information available as to how these factors are uncertain, as for example on the different rice cultivar practices), and uncertainty due to the 'fast track', i.e. applying trends to estimate latest year's emissions.

The EFs and AD uncertainties are reported in Table B2.

*Table B2. Uncertainty assigned to activity data (AD) and emission factors (EF) for CH$_4$ and N$_2$O. The table is mostly derived by IPCC GLs (IPCC, 2006) for Tier 1 emission factors, complemented with estimates by Olivier et al, (2002) and expert judgement.*

| Source category | EDGAR v4.3.2 code | | Uncertainty components | |
|---|---|---|---|---|
| | | | Uncertainty AD $u_{AD}$ (%) | Uncertainty EF $u_{EF}$ (%) |
| **CH$_4$** | | | | |
| Enteric fermentation | ENF | I<br>D<br>CS | 20 | 30<br>50<br>20 |
| Manure management | MNM | I<br>D<br>CS | 20 | 30<br>30<br>20 |
| Rice cultivation | AGS.RIC | I<br>D | 5<br>10 | [-38;+69] on default emission factors plus uncertainty on scaling factors for water regimes:<br>IRR: [-20; 26]; UPL: 0%; RNF and DWE: [-22; +26] |
| Biomass burning of crops | AWB.CRP | I<br>D<br>CS | 5<br>10<br>5 | 50<br>150<br>50 |
| **N$_2$O** | | | | |
| Manure management | MNM | I<br>D<br>CS | 20 | 50<br>100<br>50 |
| Synthetic Fertilizers; Animal Manure Applied to Soils; Crop Residue; Pasture | Direct N$_2$O emission from managed soils | I<br>D<br>CS | 20 | 70 (65 for pasture)<br>200<br>70 |
| | Indirect N$_2$O managed soils | I<br>D<br>CS | 50 | 70<br>200<br>70 |

| | Indirect | I | | 75 |
|---|---|---|---|---|
| | N$_2$O manure | D | 50 | 150 |
| | management | CS | | 75 |

**I**: industrialised (developed) countries

**D**: developing countries

**CS**: country specific

A log-normal probability distribution function is assumed to avoid negative values, and uncertainties are reported as 95 % confidence interval according to IPCC (2006, chapter 3, equation 3.7). For emission uncertainty in the range 50 % to 230% a correction factor is adopted as suggested by Frey et al (2003) and IPCC (2006, chapter 3 Uncertainties, equation 3.4). The correction factor is used as an empirical adjustment, based on Monte Carlo simulations, to correct for the deviation introduced by using the "standard" uncertainty calculation method suggested by IPCC error propagation which is only a first order approximation; for large uncertainties (as they accumulate in the propagation chain) the method systematically underestimates the uncertainty half range.

**CAPRI**

CAPRI is an economic, partial equilibrium model for the agricultural sector, focused on the EU (as well as less detailed Worldwide for market module) (Britz and Witzke, 2014[17]; Weiss and Leip, 2012[18]). CAPRI stands for 'Common Agricultural Policy Regionalised Impact analysis', and the name hints at the main objective of the system: assessing the effect of Common Agriculture Policy (CAP) instruments not only at the EU or Member State level but at sub-national level. The model is calibrated for the base year (currently 2012) and then baseline projections are built, allowing the ex-ante evaluation of agricultural policies and trade policies on production, income, markets, trade and the environment. It runs at yearly time steep at NUTS2 resolution, but can be downscaled at HSU level (approx. 1km x 1km). A detailed description can be found at: https://ec.europa.eu/clima/sites/clima/files/strategies/analysis/models/docs/capri_model_methodology_en.pdf

Among other environmental indicators, CAPRI simulates CH$_4$ and N$_2$O emissions from agricultural production activities (enteric fermentation, manure management, rice cultivation, agricultural soils). Activity data is mainly based on FAOSTAT and EUROSTAT statistics and estimation of emissions follows IPCC 2006 methodologies, with a higher or lower level of detail depending on the importance of the emission source. Details on CAPRI methodology for emissions calculations is referenced in Table A1a.

**FAOSTAT**

FAOSTAT: Statistics Division of the Food and Agricultural Organisation of the United Nations, CO$_2$, CH$_4$ and N$_2$O emissions from agriculture and LULUCF statistics till 2017, available at: http://www.FAOSTAT.org/FAOSTAT/en/#home. The FAOSTAT emissions database is computed following Tier 1

---

[17] https://www.capri-model.org/docs/CAPRI_documentation.pdf

[18] https://www.sciencedirect.com/science/article/pii/S0167880911004415

IPCC 2006 GLs for National GHG Inventories (http://www.ipcc-nggip.iges.or.jp/public/2006gl/index.html). Country reports to FAO on crops, livestock and agriculture use of fertilizers are the source of activity data. Forest data are those reported to FAO within the FRA process. Geospatial data are the source of AD for the estimates from cultivation of organic soils, biomass and peat fires. GHG emissions are provided by country, regions and special groups, with global coverage, relative to the period 1961-present (with annual updates) and with projections for 2030 and 2050, expressed as Gg $CO_2$ and $CO_2$-eq (from $CH_4$ and $N_2O$), by underlying agricultural emission sub-domain and by aggregate (agriculture total, agriculture total plus energy, agricultural soils). Similarly, "Land use total contains all GHG emissions and removals produced in the different Land use sub-domains, representing the three IPCC Land use categories: cropland, forest land, and grassland, collectively called emissions/removals from the LULUCF sector. LULUCF emissions consist of $CO_2$ (carbon dioxide), $CH_4$ (methane) and $N_2O$ (nitrous oxide) associated with land management activities. $CO_2$ emissions/removals are derived from estimated net carbon stock changes in above and below-ground biomass pools of forest land, including forest land converted to other land uses. $CH_4$ and $N_2O$, and additional $CO_2$ emissions are estimated for fires and drainage of organic soils." (http://www.fao.org/faostat/en/#data/GL/metadata).

**GAINS**

The Greenhouse gas and Air pollution Interactions and Synergies (GAINS) model (http://gains.iiasa.ac.at/) provides a framework for assessing strategies that reduce future emissions of multiple air pollutants and greenhouse gases at least costs, and minimize their negative effects on human health, ecosystems and climate change. Although the focus of GAINS is more on future scenarios and air quality policies, GAINS estimates for its baseline historical emissions from 1990 to 2050 of 10 air pollutants and 6 GHGs for each country based on data from international energy and industrial statistics, emission inventories and on data supplied by countries themselves. It assesses emissions on a medium-term time horizon, with projections being specified in five-year intervals through the year 2050 for $N_2O$ and at yearly time step for $CH_4$ (http://www.iiasa.ac.at/web/home/research/researchPrograms/air/GAINS.html). An important objective of the GAINS model is to use a consistent emission estimation methodology across all countries and sectors. Country- and sector/technology- specific emission factors are often derived in a consistent manner and are known to influence emissions, thereby producing emission estimates that are comparable across geographic and temporal scales and for which it is possible to explain deviations in emissions. By identifying the impact on emissions from implementation of various control technologies, the GAINS model can assess the expected impact on emissions from introducing additional control in the future.

**CBM**

The Carbon Budget Model developed by the Canadian Forest Service (CBM-CFS3), can simulate the historical and future stand- and landscape-level C dynamics under different scenarios of harvest and natural disturbances (fires, storms), according to the standards described by the IPCC (Kurz et al., 2009), under annual time step. Since 2009, the CBM has been tested and validated by the Joint Research Centre of the European Commission (JRC), and adapted to the European forests. It is currently applied to 26 EU MS, both at country and NUTS2 level (Pilli et al., 2016).

Based on the model framework, each stand is described by area, age and land use classes and up to 10 classifiers based on administrative and ecological information and on silvicultural parameters (such as forest composition and management strategy). A set of yield tables define the merchantable volume production for each species while species-specific allometric equations convert merchantable volume production into aboveground biomass at stand-level. At the end of each year the model provides data on the net primary production (NPP), carbon stocks and fluxes, as the annual C transfers between pools and to the forest product sector.

The model can support policy anticipation, formulation and evaluation under the LULUCF sector, and it is used to estimate the current and future forest C dynamics, both as a verification tool (i.e. to compare the results with the estimates provided by other models) and to support the EU legislation on the LULUCF sector (Grassi et al., 2018a). In the biomass sector, the CBM can be used in combination with other models, to estimate the maximum wood potential and the forest C dynamic under different assumptions of harvest and land use change (Jonsson et al., 2018).

**EFISCEN**

The European Forest Information SCENario Model (EFISCEN) is a large-scale forest model that projects forest resource development on regional to European scale. The model uses national forest inventory data as a main source of input to describe the current structure and composition of European forest resources. The model runs for five-year interval emission projections and projects the development of forest resources, based on scenarios for policy, management strategies and climate change impacts. With the help of biomass expansion factors, stem wood volume is converted into whole-tree biomass and subsequently to whole tree carbon stocks. Information on litter fall rates, felling residues and natural mortality is used as input into the soil module YASSO (Liski et al. 2005), which is dynamically linked to EFISCEN and delivers information on forest soil carbon stocks. The core of the EFISCEN model was developed by Prof. Ola Sallnäs at the Swedish Agricultural University (Sallnäs 1990). It has been applied to European countries in many studies since then, dealing with a diversity of forest resource and policy aspects. A detailed model description is given by Verkerk et al. (2016), with online information on availability and documentation of EFISCEN at http://efiscen.efi.int. The model and its source code are freely available, distributed under the GNU General Public License conditions (www.gnu.org/licenses/gpl-3.0.html).

**DGVMs (TRENDY v6)**

This study uses the ensemble of eight DGVMs that participated in TRENDY version 6 (v6) for the GCB 2017 (Le Quéré et al., 2018) including the following models: ORCHIDEE (Krinner, G. et al. 2005), OCN (Zaehle, S. et al. 2011), JULES (Clark, D. B. et al. 2011), JSBACH (Reick, C. H. et al., 2013), VEGAS (Zeng, N., 2003, 2005), LPX-Bern (Lienert and Joos 2018), LPJ (Sitch, S. 2003), ISAM (Jain, A. K. et al., 2013). We make use of carbon trends in net land carbon exchange over Europe, during the period 1990-2016. Data available for download at http://dgvm.ceh.ac.uk/index.html. DGVM models are forced by historical agricultural land cover change, climate change and $CO_2$ since 1901. The models calculate forest area from agricultural land in different ways, and, therefore can have quite very different forest areas in the EU. Models include biomass and soil C loss or gains associated with land cover change and wood harvest (diagnosed from factorial simulations) but they do not include a realistic

representation of cropland management for Europe, nor of forestry and grassland management. The time step of each of the models is described in detailed in Le Quéré et al. (2018) Table 4 and references therein.

**Bookkeeping models**

The LULUCF chapter makes use of data from two bookkeeping models: **H&N** (Houghton & Nassikas, 2017) and **BLUE** (Hansis et al., 2015). As described by GCB 2017 (Le Quéré et al., 2018): "H&N model (Houghton, 1983) calculate land use change $CO_2$ emissions and uptake fluxes for transitions between various natural vegetation types and agricultural lands (croplands and pastures). The original bookkeeping approach of Houghton (2003) keeps track of the carbon stored in vegetation and soils before and after the land use change. Carbon gain or loss is based on

response curves derived from literature. The response curves describe decay of vegetation and soil carbon, including transfer to product pools of different life-times, as well as carbon uptake due to regrowth of vegetation and consequent re-filling of soil carbon pools. Natural vegetation can generally be distinguished into primary and secondary land. For forests, a primary forest that is cleared cannot recover back to its original carbon density. Instead long- term degradation of primary forest is assumed and represented by lowered standing vegetation and soil carbon stocks in the

secondary forests. Apart from land use transitions between different types of vegetation cover, forest management practices in the form of wood harvest volumes are included. Different from dynamic global vegetation models, bookkeeping models ignore changes in environmental conditions (climate, atmospheric $CO_2$, nitrogen deposition and other environmental factors). Carbon densities at a given point in time are only influenced by the land use history, but not by the preceding changes in the environmental state. Carbon densities are taken from observations in the literature

and thus reflect environmental conditions of the last decades".

       The **BLUE** model provides a data-driven estimate of the net land use change fluxes. BLUE stands for 'bookkeeping of land use emissions'. Bookkeeping models (Hansis 2015, Houghton 1983) calculate land use change $CO_2$ emissions (sources and sinks) for transitions between various natural vegetation types and agricultural lands. The bookkeeping approaches keep track of the carbon stored in vegetation, soils, and products before and after the land

use change.

       In BLUE, land use forcing is taken from the Land Use Harmonization, LUH2, for estimates within the annual global carbon budget. The model provides data at annual time steps and 0.25 degree resolution. Temporal evolution of carbon gain or loss, i.e. how fast carbon pools decay or regrow following a land use change, is based on response curves derived from literature. The response curves describe decay of vegetation and soil carbon, including transfer

to product pools of different lifetimes, as well as carbon uptake due to regrowth of vegetation and subsequent refilling of soil carbon pools.

**Wetland emissions ensemble of models**

       This model ensemble simulates natural $CH_4$ emissions from wetlands and contains eleven biogeochemical models

(CLM4.5 (Riley et al., 2011), CTEM, DLEM (Tian et al., 2010), VISIT (Ito and Inatomi 2012), JULES (Hayman et al., 2014), LPJ-MPI (Kleinen at al., 2012), LPJ-wsl (Hodson et al., 2011), LPX-Bern (Spahni et al., 2011), ORCHIDEE (Ringeval et al., 2010), SDGVM (Hopcroft et al., 2011), TRIPLEX-GHG (Zhu et al., 2015)). These models are

referenced and can be found in Poulter et al., 2017 Supplementary Information: https://iopscience.iop.org/1748-9326/12/9/094013/media/ERL_12_9_094013_suppdata.pdf

**Appendix C**

**Example of country specific uncertainty calculation for LULUCF sector 4**

Table C1: Aggregation of IPCC sub-sectors for the uncertainty analysis

| Energy Sector (CRF 1A) | 1A1, 1A2, 1A3, 1A4, 1A5 |
|---|---|
| Fugitive Emissions Sector (CRF 1B) | 1B1, 1B2 |
| IPPU Sector (CRF 2) | 2A, 2B, 2C, 2D, 2E, 2F, 2G, 2H |
| Agriculture Sector (CRF 3) | 3A, 3B, 3C, 3D, 3E, 3F, 3G, 3H |
| LULUCF Sector (CRF 4) | 4A, 4B, 4C, 4D, 4E, 4F, 4G, 4H |
| Waste Sector (CRF 5) | 5A, 5B, 5C, 5D, 5E |


For a better understanding and overview of the single steps of the Uncertainty Analysis, an example calculation for Uncertainty Assessment is included, where the combined uncertainty and contribution to variance is calculated for 4A $CO_2$. The same was done for 4B, 4C etc.

1. Table C2 shows the subsectors 4A and 4B of one the EU28 MS Uncertainty Assessment for 2016.

Table C2: Calculation example of the uncertainty analysis; uncertainty assessment 2016.

| IPCC Code | Gas | Year $x$ emissions or removals | Combined uncertainty | Contribution to Variance by Category in Year $x$ |
|---|---|---|---|---|
| | | Input data | $\dfrac{(G*D)^2}{(\sum D)^2}$ | $\sqrt{E^2 + F^2}$ |
| | | kt $CO_2$ equivalent | % | |
| 4 A | CH4 | 237,955502 | 0,24758837 | 0,00 |
| 4 A | CH4 | 40,1939139 | 1,06066017 | 0,00 |
| 4 A | CO2 | -30251,343 | 0,24758837 | 0,00 |
| 4 A | CO2 | -5829,38043 | 1,06066017 | 0,00 |
| 4 A | N2O | 0,8914493 | 0,24758837 | 0,00 |
| 4 A | N2O | 0,15057789 | 1,06066017 | 0,00 |
| 4 B | CH4 | 2,05470896 | 1,06066017 | 0,00 |
| 4 B | CO2 | 1917,59719 | 1,06066017 | 0,00 |
| 4 B | CO2 | 541,959929 | 1,06066017 | 0,00 |
| 4 B | N2O | 0,76975268 | 1,06066017 | 0,00 |
| 4 B | N2O | 26,874189 | 1,06066017 | 0,00 |
| Total emissions of all sectors | | 397935,125 | | |

2. To calculate the contribution to variance for the sector 4A $CO_2$, the following steps have to be performed:

    (1) (-30251,343) + (-5829,38) = (- 36080,72) (building the sum of the emissions of year x for 4A, $CO_2$)

    (2) ((-30251,343 * 0,24758837)^2 + (-5829,38 * 1,06066017)^2) / (-36080,72)^2 = 0,0724584 (intermediate step for calculating the Combined Uncertainty)

    (3) SQRT(0,0724584) = 26,918 % (Combined Uncertainty)

    (4) ((-36080,7234* 26,918)^2) / 397935,125^2 = 0,001 (Contribution to Variance for year x)

3. Results can be found in table C3

Table C3: Calculation example of the uncertainty analysis; section from one of the MS of the EU28 uncertainty assessment 2016.

| IPCC Code | Gas | Year x emissions or removals | Combined uncertainty | Contribution to Variance by Category in Year x |
|---|---|---|---|---|
| | | Input data | $\frac{(G*D)^2}{(\sum D)^2}$ | $\sqrt{E^2 + F^2}$ |
| | | kt $CO_2$ equivalent | % | |
| 4A | CO2 | -36080,7234 | 0,26918099 | 0,001 |

To check for correctness, the total uncertainty for the aggregated sectors can be calculated. If the total uncertainty for the aggregated sectors matches the total uncertainty of the uncertainty assessment, the calculated uncertainties for the subsectors are correct. This was the case for all calculations performed for this analysis.

The results of the Uncertainty Analysis show a clear trend of the main uncertainties and gases across the analyzed 26 EU MS.

**Appendix D**

**Country specific emissions**

Detailed agriculture $CH_4$ and $N_2O$ emissions split in activities for all EU28 countries can be downloaded at the following link: https://zenodo.org/record/3662371#.Xkui-WhKjIU and are found under the "Figures5,8_AppendixD_$CH_4$_$N_2O$_per_country_new" excel document.

**Data availability**

All raw data files reported in this work which were used for calculations and figures are available for public download at https://zenodo.org/record/3662371#.Xkui-WhKjIU (Petrescu et al., 2020).The data we submitted is reachable with one click (without the need for entering login and password), and a second click to download the data,

consistent with the two-click access principle for data published in ESSD (Carlson and Oda, 2018). The data and the DOI number is subject to future updates and it refers only to this version of the manuscript.

**Acronyms and abbreviations**


AD Activity data

AFOLU Agriculture, Forestry and Other Land use

AR Assessment Report

BP The British Petroleum Company

CAPRI Common Agricultural Policy Regionalised Impact analysis model

CBM Carbon Budget Model

$CH_4$ Methane

$CO_2$ Carbon dioxide

CAP Common Agriculture Policy

CLM4.5 Community Land Model

COP Conference of the Parties

CRF Common Reporting Format

CTEM The Canadian terrestrial ecosystem model

DG CLIMA Directorate General CLIMA (European Commission)

DGVMs (TRENDY) Dynamic global vegetation models

DLEM Dynamic land ecosystem model

EDGAR Emission Database for Global Atmospheric Research

EEA European Environmental Agency

EF Emission factor

EFISCEN European Forest Information SCENario Model

EIT Parties: Economies in Transition

ESA CCI European Space Agency Climate Change Initiative

ETS Emissions Trading System

EU28 European Union

EUROSTAT European Statistical Office

FADN Farm Accountancy Data Network

FAOSTAT Food and Agriculture Organization of the United Nations

FL Forest Land

FOLU Forestry and Other Land use

FRA Global Forest Resources Assessment

GAINS Greenhouse gas and Air pollution Interactions and Synergies model

GCP Global Carbon Project

GCB Global Carbon Budget

GHG Greenhouse Gases

GHGI Greenhous Gas Inventory

GMB Global Methane Budget

H2020 Horizon 2020

IEA International Energy Agency

IFA International fertilizer industry organization

IPCC Intergovernmental Panel on Climate Change

IPCC GLs IPCC Guidelines

IPCC SRCCL IPCC Special Report on Climate Change and Land

IPPU Industrial processes and product use

JRC Joint Research Centre of the European Commission

JULES the Joint UK Land Environment Simulator

KP Kyoto Protocol

LPJ-MPI Lund-Potsdam-Jena model - Max-Plank Institute version

LPJ-wsl Lund-Potsdam-Jena model– WSL version

LPX-Bern Land surface Processes and eXchanges" model of the University of Bern

LULUCF Land use, Land use Change and Forestry

MODIS Moderate resolution imaging spectroradiometer

MS Member States

$N_2O$ Nitrous oxide

NBP Net Biome Productivity

NBS National Bureau of Statistics of China

NDCs - Nationally Determined Contributions

NEP Net Ecosystem production

NFI National forest inventory

NGHGI National Greenhous Gas Inventory

NIRs National Inventory Reports

NPP Net Primary Production

NUTS2 Nomenclature of territorial units for statistics

OECD The Organisation for Economic Co-operation and Development

ORCHIDEE Organising Carbon and Hydrology In Dynamic Ecosystems

PA Paris Agreement

PBL Planbureau voor de Leefomgeving (Netherlands Environmental Assessment Agency)

SBSTA Subsidiary Body for Scientific and Technological Advice

SDGVM Sheffield Dynamic Global Vegetation **Model**

TACCC Transparency, Accuracy, Completeness, Comparability, Consistency

1235 TRIPLEX-GHG a hybrid, monthly time-step model of forest growth and carbon dynamics coupled with a new methane ($CH_4$) biogeochemistry module

UNEP United Nation Environment Programme

UNFCCC United Nations Framework Convention on Climate Change

USGS United States Geological Survey

1240 VERIFY Verifying greenhouse gas emissions, EU H2020 project, grant agreement No 776810

VISIT Vegetation Integrative Simulator for Trace Gases

WSA World steel association

WWII World War two

**Author contributions**

A.M.R.P and H.D. designed research and led the discussions, A.M.R.P. analyzed the data and wrote the initial version of the paper; G.P. provided the figures and initial text for the LULUCF chapter, A.M.R.P., P.C., H.D., G.P., G.G., G.J.M, F.N.T., W.W. made significant changes throughout all versions of the paper, G-J.N., A.L., G.C-G., L.H-J., E.S., R.P., A.K., A.B., J.P., G.C. and R.M.A. reviewed the initial versions of the paper and provided comments, suggestions and advice during the preparation of this manuscript., A.K. developed the methodology for the UNFCCC uncertainty calculation for each Member State, provided the information and contributed to the writing of Appendix C, E.S. developed the methodology for the EDGAR uncertainty calculation and provided the $CH_4$ and $N_2O$ uncertainties, D.G. contributed to the writing of the UNFCCC description (Appendix B), R.M.A. provided the initial text for the UNFCCC and EDGAR descriptions (Appendix B), M-J.S. R.P., G.C-G., L.H-J., W.W., A.B., G.C, J.E.M.S.N. are data providers and advised during data analysis process.

**Competing interests**

The authors declare that they have no conflict of interest.

**Acknowledgements**

The authors acknowledge funding from the European Union's Horizon 2020 research and innovation programme, VERIFY project, grant agreement No. 776810. We thank the TRENDY international modelling team for access to DGVMs v6 output and Benjamin Poulter for providing us with the data for natural $CH_4$ emissions from wetlands model ensemble. We acknowledge here all modellers of this ensemble as following: Joe Melton (CTEM), Hao Shi (DLEM), Akihiko Ito (VISIT), Nic Gedney (JULES), Changhui Peng (TRIPLEX-GHG), Shushi Peng (ORCHIDEE), Thomas Kleinen ( LPJ-MPI), Zhen Zhang (LPJ-WSL), Jurek Mueller (LPX), Roley W.J. (CLM4.5) and Hopcroft (SDGVM).

J.P was supported by the German Research Foundation's Emmy Noether Program. We thank Raul Abad Viñas for advice received during the making of Figure 4 and Simone Rossi for providing activity data (forest area and wetlands

CH$_4$ emissions) from the UNFCCC NGHGI submissions. We thank as well Xavi Rotllan Puig for his support offered during the UNFCCC NGHGI 2018 data download.

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
