# Peer review of "European anthropogenic AFOLU greenhouse gas emissions: a review and benchmark data"

_Earth System Science Data, 2019_

## Referee Comment (RC1) · Anonymous Referee #1 · 5 Jan 2020

The manuscript and the data, if published, will be very useful, in particular, for modelers who need a benchmark for assessing the performance of their models. The authors present a collection of data on greenhouse gas (GHG) emissions in the AFOLU from the EU countries' submissions to the UNFCCC, FAOSTAT, and, that is more valuable, different models. The authors provide an overview of the emissions by GHG and activity, and shortly describe the methods and models used for obtaining the emission estimates. The uncertainties are presented only for the UNFCCC data and EDGAR. All the figures in the manuscript are complemented with respective datasets in Excel tables. The manuscript is well written as an overview, however, the data are not well structured and described (the data are provided only as a support for the figures in

the manuscript, there is no detailed emission data for all activities, or I didn't find the main file with all the data...). It is difficult to understand the data without reading the manuscript. The authors did not describe how they found the data from the models and what criteria were used for selecting the models (which search websites, what keywords, publication timeframe, journals, etc.). I'd avoid using the word "benchmark" in the title, I agree, it's a good collection of the data of different origin, but I'm not convinced by the manuscript that it's a benchmark. Since the uncertainties are presented only for two data sources, I'd avoid using the word "uncertainties" in the title as well. From a publication with such title I expect, except the review, a dataset with all the data grouped by activity, GHG, sources with the best possible time and geospatial resolution; with a section (or annex) devoted to the description of the dataset. In addition, the data should be better explained in the excel table, e.g. using comments or notes. Specific comments are presented below. Line 76: some countries use different base year than 1990, e.g., Hungary. Line 77: what is year-2? Lines 150-155: Please explain better how did you search for the data, i.e., which search websites, what keywords, selection criteria etc. Table 1: Please improve the table for easier distinguishing the rows describing the GHGs. Lines 420-425: Please try to use the same units (either Tg or kton). Line 450: "the" at the beginning of the line is redundant. Figure 13: I didn't find the respective data in the Excel files. Footnote 6 on p.37: "...between then UNFCCC..." to be changed to "...between the UNFCCC...". Table B2: Please explain UAD and UEF, which confidence interval? Lines 980-990/ 1035-1045/ 1050-1060 / 1065-1075 / 1080-1090: Please include information on the timestep of the models.

---

## Referee Comment (RC2) · Anonymous Referee #2 · 18 Jan 2020

The authors present a comprehensive compilation of greenhouse gas (GHG) emissions from the agriculture, forestry and other land use (AFOLU) sector in Europe based on official country reports to UNFCCC and FAOSTAT, the global EDGAR inventory and several different types of bottom-up models. They provide an overview of the basic methodologies underlying the different emission estimates and present information on uncertainties where available. Unfortunately, uncertainty estimates are only available for UNFCCC and EDGAR. It would, however, be important to get estimates of the uncertainties for the other datasets and in particular for the models as well. The authors check for inconsistencies between the different approaches and also trace where the same underlying information is used, which might result in an overoptimistic interpretation of the consistency of emission estimates for some sectors.

Together with the review the authors also provide a compilation of the emission data themselves (in the form of excel sheets), not only those shown in the Figures but also some underlying data, like time series of emissions from different information sources. The excel sheets would be even more useful, if a better and more comprehensive documentation was included directly in the tables. This is highly recommended. In order to serve as benchmark for models and other inventories a better documentation of the emission data sets is indispensable. The excel files should also be restructured in a way that automated reading and processing is possible in programming languages like python or R.

Furthermore, a comprehensive list of all original data sources would be important. The authors should make sure that the underlying data are traceable by clear and direct reference to the original providers.

This compilation provides a good starting point for a detailed analysis of the European carbon budget but similar synthesis efforts are needed for the other components.

The main part of the paper is well written and well structured. The data source description in Appendix B, however, needs some careful editing, in particular the section describing UNFCCC, see specific comments below. The authors should also carefully check the paper for ambiguous use of the word 'source' and clearly distinguish between emission source and data/information source.

Specific comments:

Line 75: Please add a brief definition of 'Annex I countries'

Line 77: 'current year-2' - please explain in words what is meant

Line 79: add year and reference for Paris agreement

Line 80: What is CO2e? Please define before using.

Line 81: Please explain GWP100 and give reference for its definition.

Line 81: Please explain acronym LULUCF and how this relates to AFOLU.

Line 80 and 87: Is there a difference between '2018 NGHGI estimates' and 'NGHGI 2018 data'? If not, use consistent naming. Otherwise explain difference.

Figure 1: What is 'remaining land' and 'converted land' and how does this relate to AFOLU? What is HWP? Please explain in the legend or add clear references to these terms in the text.

Line 106: Please define FOLU. Is this exactly the same as LULUCF?

Line 112 and 120: Do these 'total EU28 GHG emissions' include the LULUCF sector?

Line 131: Will the comparison between bottom-up and top-down estimates of CH4 and N2O emissions be part of a following study? Better mention the top-down studies in the conclusions where you outline the next steps.

Line 152 and Table A2: In contrast to 'For all three GHGs, total emissions... in Table A2', there are no CO2 emissions listed in that specific table.

Line154: What are 'observational data streams' in this context? Is there anything measured or observed? Please clarify.

Lines 198-200: Consider moving the information on CO2 emissions from agriculture to the introduction of Section 3 as it does not fit well in a section explicitly on CH4 and N2O emissions.

Figure 2 legend: 'The positive values represent a source.' Define somewhere in the main text that source and sink, positive and negative values respectively, are defined from an atmosphere point of view.

Line 325: The disagreement in % change of emissions between last reported year and 2005 between the different data source is of course also partly due to the different

length of the time period and hence not necessarily surprising. Please comment.

Line 386: 'top-down and bottom-up SOURCES' sounds misleading, rather 'top-down and bottom-up estimates' or similar. Are these 'natural CH4 emissions' ? Probably not, please clarify.

Line 466: Please clarify this part of the sentence: 'carbon emissions from fire (some models and CO2 emissions from harvested wood products)'

Line 499: Please add an explanation or reference why removing af/reforestation from FAOSTAT numbers would result in an even bigger sink?

Line 516: Please explain acronym FAO-FRA. What is the difference to FAOSTAT mentioned in the previous sentence?

Line 543-546: Not entirely clear how much of Finland and Sweden is covered by grassland: ' most of Finland and Sweden' or ' grassland covered less than 6 % of the land in Finland and Sweden'.

Line 557: Can the increase of emissions from grasslands in 2016 in UNFCCC be explained by changes in specific countries? This would be worth mentioning.

Line 625: Is 'Houghton's estimate' the same as the H&N results? If so, please use consistent names.

Figure 13: The underlying data for this figure seem to be missing. At least I could not find them in any excel file. Is there a reason why these data are not provided? If so please comment in the paper.

Figure 13: What is the meaning of 'Model (Cumulative from 1990)'?

Line 676: The link points to the JRC report of the workshop on 'Atmospheric monitoring and inverse modelling for verification of greenhouse gas inventories' (Bergamaschi et al., 2018) and not to an InGOS report. Please check and provide correct citation.

Line 728: Why is freshwater CO2 emission considered to be a disturbance?

Figure 14 legend: The explanation of what is included in the DGVMs and how changes are derived needs some clarification, it is difficult to follow.

Line 811: It is true that global inversions are needed for a consistent constraint of the global carbon balance but for an estimate of the European carbon fluxes from inverse modelling, high-resolution regional inversion systems would be more appropriate, of course coupled with global inversions. The regional systems are supposed to better represent atmospheric transport and spatial heterogeneity of the fluxes. VERIFY will also make use of regional inversions.

Line 815: What is meant by 'total emission column'?

Line 815 ff: Also regional inversions have been used to estimate European CH4 and N2O emissions (Bergamaschi et al., 2015; Bergamaschi et al., 2018). Might be worth mentioning.

Appendix B UNFCCC: Please give a list, incl. references, of all UNFCCC guidelines in chronological order and refer to them in the appendix text and also in the main text of the manuscript to provide a clearer structure and better traceability of the statements. In the current state this section is rather unstructured and difficult to follow.

Line 884 ff: These are the rules to combine uncertainties using error propagation not to compute uncertainties in general. You do not mention the two different approaches to combine uncertainties, i.e. error propagation and Monte Carlo simulation. Is the Monte Carlo method not used or not relevant?

Line 894: Table B1 is interesting to know but which sectors are missing?

Lines 909-912: What is the difference to lines 867-871?

Lines 913ff: What is the difference between this paragraph and the information in the paragraphs starting line 851 and 865, respectively?

Please restructure the whole section describing UNFCCC

Line 977: What type of correction factor is this? What is corrected and why? Please specify.

Technical corrections:

Line 61: DGVM - please explain acronyms when they first appear, in addition to the acronym list.

Line 87 and Figure 1: Use consistent naming CO2e or CO2-eq

Line 129: ... top-down GHG emission estimates.

Table 1: To enhance readability please structure the table in a way that the different gases are highlighted and that for all gases the same dataset is in the same column.

Table 2: To enhance readability please format the table in a way that there is one row per sector and combined rows where needed.

Figures 2, 5, 6, 10 legends: Use same name for a data source. Or are 'UNFCCC 2018 communications', 'UNFCCC NGHGI 2018', 'UNFCCC' different data sources?

Line 345: Reference to dataset was already given above in the same paragraph, remove here.

Line 345: Please clarify to what 'count for as much as ...' refers to. Consistency? Agriculture emissions from the countries listed?

Line 346: 'reported by UNFCCC' or 'reported to UNFCCC'?

Line 394: depends -> depend

Line 395: ...estimate to have...

Line 425: The acronym GHGI is missing in acronym list.

Line 450: separate -> separately

Line 451: because, -> , because

Line 457: Please clarify the structure of the sentence.

Line 576: Something seems to be missing in 'that is, the sink that favorable environmental changes, in particular CO2 fertilization'

Figure 12: Shows also definition of anthropogenic land in DGVMs, please include in the legend.

Figure 12 legend: Add reference to specific version of GCP.

Line 666: remove 'takes'

Line 685: According to...

Line 730: 'the last GCP 2018'. Please remove 'last' as there is no other GCP2018 and it is no longer the last GCP.

Line 741: ' In order to limit temperature increase to 1.5°C and keep it below 2°C' does not sound logical, please rephrase.

Line 768: '*changes*' Is there any meaning in the starts?

Line 783: countries reports -> county reports

Line 788: Please specify what is flowing... e.g. information flow between...

Line 819: Please give proper reference of InGOS report - if that is what you refer to.

Line 899: 'level uncertainties'?

Line 983: Explain acronym CAP and add to acronym list.

References:

Bergamaschi et al.: Atmospheric monitoring and inverse modelling for verification of greenhouse gas inventories, EUR 29276 EN, Publications Office of the European Union, Luxembourg, 2018, ISBN 978-92-79-88938-7, doi:10.2760/759928, JRC111789

Bergamaschi,et al.: Top-down estimates of European CH4 and N2O emissions based on four different inverse models, Atmos. Chem. Phys., 15, 715–736, https://doi.org/10.5194/acp-15-715-2015, 2015.

Bergamaschi et al.: Inverse modelling of European CH4 emissions during 2006–2012 using different inverse models and reassessed atmospheric observations, Atmos. Chem. Phys., 18, 901–920, https://doi.org/10.5194/acp-18-901-2018, 2018.

---

## Author Response (AR1)

**Dear Topical Editor David Carlson,**

**Dear Reviewers and Editorial Board of ESSD,**

As requested, we submit a list of point-by-point relevant changes which supplement the answers already submitted and included below, answers to reviewers' comments.

We attach a yellow marked-up version of the manuscript. We will not refer here to grammar or language corrections, but they appear in the marked-up manuscript. The lines refer to the marked-up version.

Line 1: As already mentioned in our responses, the title is: "European anthropogenic AFOLU greenhouse gas emissions: a review and benchmark data"

Line 26: Added footnote 1 explaining the EU28 country composition

Lines 39-46: new zenodo link was updated and new citation added

Line 59: DGVM acronym is explained

Lines 68-70: we rephrase the paragraph

Figure 1 caption was re-written adding explanation and footnote 6 explaining the management classes

Lines 115-120: We reformulated and added the correct % for incl. and excl. LULUCF

Line 150: we added footnote 7 for the link of the EU LULUCF regulation.

Line 162: we added "and reported"

Table 1: $CH_4$ we added new reference Höglund-Isaksson, L. 2020

Line 174-177: the zenodo link and reference were updated

Line 206: We use GLs for Guidelines throughout the whole paper

Line 310: we added "(1989 and 1988 respectively)" to explain the different base years of Romania and Bulgaria

Table 3: we added "minus 3.B.2.5" which was omitted in the previous version

Line 401 and 403-404: we reformulated the sentences

Line 406: we added the web link under footnote 8

Line 416: we removed the web link and referred to footnote 7

Figure 9 caption: we added "The models are explained in the acronym list and referenced in Appendix B"

Line 447: we reformulated the sentence

Line 455: we added to the sentence more option for fulfilling PA requirements and reaching the well-below $2^o$ goal

Line 476-479: EFISCEN rephrased and updated their NBP definition

Line 481-482: DGVMs definition was updated

Table 4: FAOSTAT: added 3 references; H&N added explanation on inclusion of estimates from peat burning and peat drainage

Lines 508-510: we added explanation on FAOSTAT estimates (i.e. afforestation)

Lines 513-515: small language corrections

Figure 10: we changed the colors to match the other figures 2,3,6,7 (e.g. FAOSTAT orange, UNFCCC dark blue)

Line 525-535: we re-wrote the paragraph and added values for the sinks and differences (Finland, Romania, Latvia and Denmark)

Lines 561-565: we rephrased to allow flow to the text and refer to the EUROSTAT link in the reference list

Figure 12: we added extra explanation to the caption

Line 621: We added reference to IPCCC 2010

Line 624-629: we removed the explanation on land use proxy IPCC "

 "

as already present and discussed in Figure 12.

Line 642-645: we rephrased the paragraph

Line 656-659: we detailed the H&N use of input data

Figure 13: We changed the colours to match the other figures 2,3,6,7,10 and data sources (FAOSTAT orange, UNFCCC dark blue)

Line 743: rephrased sentence on practice guidance

Figure 14: we replaced the old figure with this one, conceptual but which tries to quantify the land components for EU28. Figure caption was as well re-written

Lines 840-845: we added sentences on the European Commission's decision to include a service for monitoring anthropogenic $CO_2$ emissions under Copernicus program.

**Appendices**

Line 1016 FAOSTAT description was updated by the data providers

Line 1105: BLUE description was updated by the data providers

The remaining marked-up line-by-line corrections from Appendices are included in the submitted responses, you kindly find below.

**Acronyms**

We added missing acronyms belonging to CH4 models (section 3.2)

**References**

We added some references as kindly suggested by reviewers, and not only.

**Interactive comment on:**

**European anthropogenic AFOLU emissions and their uncertainties: a review and benchmark data" by A.M.R. Petrescu et al.**

**REPLY TO THE REFEREE #1**

The authors thank Referee #1 for the thoughtful and helpful comments and for the fact that the reviewer acknowledges the manuscript as being a comprehensive collection of data, very useful for the modelers and the whole scientific community. In the revised version we implemented the suggestions regarding the structure of the metadata and information on data sources.

**General evaluation:**

This study is intended to be annually updated, similar to the GCP papers (Friedlingstein et al., 2019), to evolve into a complete synthesis of bottom-up and top-down GHG estimates of European countries and ecosystems. While the GCP provides the global carbon budget, this study starts a series of datasets for EU. These are essential for the GHG Monitoring and Verification Support (MVS) capacity, the EU envisages to build in support of the enhanced transparency framework of the Paris Agreement. The European Commission decided to take up under the long-term Copernicus a new service for monitoring anthropogenic $CO_2$ emissions, which is under construction (Janssens-Maenhout et al., 2020).

While quantitative estimates of uncertainties are, at this stage in the project, only available for UNFCCC and EDGARv4.3.2, we agree that more is needed to evaluate the validity of the model results and between various data sources. We will, therefore remove the word "uncertainties" from the title: ***European anthropogenic AFOLU greenhouse gas emissions: a review and benchmark data***

Regarding the wording and the use of "benchmark" term in the title, we strongly believe that this compilation of available data extends significantly beyond what was presented by any other publication, for AFOLU and also for Europe. Therefore, any future quantification of AFOLU GHGs in Europe will need to consult this dataset as a benchmark to compare their own result to – especially given our intent to continuously update. Similar exercises in other world regions could also use this as a valuable basis for comparison. Hence we propose to keep the word benchmark in the title.

The data uploaded on the initial zenodo link http://doi.org/doi:10.5281/zenodo.3460311 represents the data behind the figures, it ensures easy replicability. We updated the link with https://zenodo.org/record/3662371#.Xkui-WhKjIU and, next to updated figure files, we added as well the original files "metadata_" of public databases (EDGAR v4.3.2., FAOSTAT and UNFCCC NGHGI 2018). CAPRI and CBM original time series are provided as well. For the rest of the data, the co-authors would prefer to be first contacted in line with their data policy (see contact details in Appendix A, Tables 1A1 and 1A2 extra column providing the Emission Data availability) before providing their full times series (H&N, EFISCEN, GAINS, TRENDY v6, BLUE).

Regarding the description of the datasets, this is located in Appendix B, where we describe each source and where we added information on spatial resolution, time steps and we updated as well the data in the excel sheets to ensure a better readability as suggested by the reviewer.

**Response to specific comments and changes in manuscript:**

We attached online to the response the revised manuscript highlighting in yellow all the changes as responses to both reviews. Please note that the line numbers changed. We refer below to the line numbering from the first submission version you kindly reviewed.

Line 76: yes, it is true that the base year differs for some countries, therefore we added a footnote no. 3 with the following explanation: *For most Annex I Parties, the historical base year is 1990. However, Parties included in Annex I with an Economy in Transition during the early 1990s (EIT Parties) were allowed to choose one year up to few years before 1990 as reference because of a non-representative collapse during the breakup of the Soviet Union (e.g. Bulgaria (1988), Hungary (1985-87), Poland (1988), Romania (1989) and Slovenia (1986)).*

Line 77: we removed the term year-2 from the main text

Line 150-155: We added a sentence in the manuscript Section 2, line 150, stating that the reason for choosing these datasets was: *"The collection of data represents the latest data available and most recent state of the art of available estimates of GHGs representing the AFOLU sector in Europe as derived from our knowledge of the scientific literature, the scientific networks in Europe."* We explain as well in the Conclusions, that the following synthesis will include data produced under the VERIFY H2020 project, where most of the modeled data presented here, will be again analyzed. We expect this to continue also after the VERIFY project has formally ended into the Copernicus $CO_2$ service that is currently developed (Janssens-Maenhout et al., 2020).

Table 1: We added a line for each gas to better separate between the three sections.

Line 420-425: Thanks for reminding us of this inconsistency. We changed "kton $CH_4$" into Mg CH4. In brackets we added the Tg corresponding value.

Line 450: removed "And"

Figure 13: on the new zenodo link we added the excel sheet with data belonging to Figure 13 as well as Figure 1.

Footnote 6 (now 13): is corrected

Table B2: We inserted the explanation for $u_{AD}$ and $u_{EF}$. The confidence interval is 95% and is mentioned in the paragraph below Table B2.

Appendix B data source descriptions: we added, whenever available, information on the time step of the models.

*Friedlingstein, P., Jones, M. W., O'Sullivan, M., Andrew, R. M., Hauck, J., Peters, G. P., Peters, W., Pongratz, J., Sitch, S., Le Quéré, C., Bakker, D. C. E., Canadell, J. G., Ciais, P., Jackson, R. B., Anthoni, P., Barbero, L., Bastos, A., Bastrikov, V., Becker, M., Bopp, L., Buitenhuis, E., Chandra, N., Chevallier, F., Chini, L. P., Currie, K. I., Feely, R. A., Gehlen, M., Gilfillan, D., Gkritzalis, T., Goll, D. S., Gruber, N., Gutekunst, S., Harris, I., Haverd, V., Houghton, R. A., Hurtt, G., Ilyina, T., Jain, A. K., Joetzjer, E., Kaplan, J. O., Kato, E., Klein Goldewijk, K., Korsbakken, J. I., Landschützer, P., Lauvset, S. K., Lefèvre, N., Lenton, A., Lienert, S., Lombardozzi, D., Marland, G., McGuire, P. C., Melton, J. R., Metzl, N., Munro, D. R., Nabel, J. E. M. S., Nakaoka, S.-I., Neill, C., Omar, A. M., Ono, T., Peregon, A., Pierrot, D., Poulter, B., Rehder, G., Resplandy, L., Robertson, E., Rödenbeck, C., Séférian, R., Schwinger, J., Smith, N., Tans, P. P., Tian, H., Tilbrook, B., Tubiello, F. N., van der Werf, G. R., Wiltshire, A. J., and Zaehle, S.: Global Carbon Budget 2019, Earth Syst. Sci. Data, 11, 1783–1838, https://doi.org/10.5194/essd-11-1783-2019, 2019.*

*Janssens-Maenhout, G., Pinty, B., Dowell, M., Zunker, H., Andersson, E., Balsamo, G., Bézy, J.-L., Brunhes, T., Bösch, H., Bojkov, B., Brunner, D., Buchwitz, M., Crisp, D., Ciais, P., Counet, P., Dee, D., Denier van der Gon, H., Dolman, H., Drinkwater, M., Dubovik, O., Engelen, R., Fehr, T., Fernandez, V., Heimann, M., Holmlund, K., Houweling, S., Husband,*

*R., Juvyns, O., Kentarchos, A., Landgraf, J., Lang, R., Löscher, A., Marshall, J., Meijer, Y., Nakajima, M., Palmer, P., Peylin, P., Rayner, P., Scholze, M., Sierk, B., Veefkind, P., Towards an operational anthropogenic $CO_2$ emissions monitoring and verification support capacity, accepted for publication in BAMS, doi:10.1175/BAMS-D-19-0017.1, 2020 forthcoming*

**Interactive comment on:**

**European anthropogenic AFOLU emissions and their uncertainties: a review and benchmark data" by A.M.R. Petrescu et al.**

**REPLY TO THE REFEREE #2**

The authors thank Referee #2 for the thoughtful, helpful and very detailed comments and for the fact that they acknowledged the manuscript as being a comprehensive compilation of GHG AFOLU data. In the revised version we implemented the reviewer's suggestions regarding the structure, metadata and specific technical comments which were well appreciated.

**General evaluation:**

As mentioned above in our response to Reviewer 1, this study is intended to be annually updated, similar to the GCP papers, to become a complete synthesis of bottom-up and top-down GHG estimates of European countries and ecosystems.

While the GCP provides the global carbon budget, this study starts a series of datasets for EU. These are essential for the GHG Monitoring and Verification Support (MVS) capacity, the EU envisages to build in support of the enhanced transparency framework of the Paris Agreement. The European Commission decided to take up under the long-term Copernicus a new service for monitoring anthropogenic $CO_2$ emissions, which is under construction (Janssens-Maenhout et al., 2020).

As the referee commented, the other components (e.g. $CO_2$ emissions from fossil fuels) and introduction of inverse estimates will be soon finalized in an updated synthesis of this current study.

Indeed, uncertainties are, at this stage in the project, only available for UNFCCC and EDGARv4.3.2 and we agree to the fact that more is needed to evaluate the validity of model results and various data sources. As also suggested by Referee #1, We will, therefore remove the word "uncertainties" from the title: ***European anthropogenic AFOLU greenhouse gas emissions: a review and benchmark data***

Regarding the more comprehensive documentation for the excel tables, we added in Appendix A, Tables 1A1 and 1A2 an extra column providing the Emission Data availability (download links or contact persons). In order to read the data using different programming language, we would advise to use the original downloaded time series (as described in Tables 1A1 and 1A2). For data policy purposes we cannot provide all these detailed data for all sources. EDGAR v4.3.2., FAOSTAT and UNFCCC NGHGI 2018 are public databases therefore we added the original files "metadata_" to the new zenodo link https://zenodo.org/record/3662371#.Xkui-WhKjIU. CAPRI and CBM original time series were as well uploaded. For the rest of the data the co-authors would prefer to be first contacted before providing their full times series, in line with their data policy (H&N, EFISCEN, GAINS, TRENDY v6, BLUE).

We took on board the comment on re-writing the Appendix B UNFCCC description, therefore, the new version of the manuscript includes a more comprehensive and more consistent text, therefore, the specific line by line responses do not include those belonging to the old UNFCCC description.

**Response to specific comments and changes in manuscript:**

We attached online, next to the responses, the revised manuscript highlighting in yellow all edits in track changes as well as a clean version which ensures a better readability. Please note that the line numbers changed. We refer below to the line numbering from the first submission version you kindly reviewed.

Line 75: We added the footnote no. 2 with Annex I definition

Line 77: We removed the year-2 terminology from the text

Line 79: We added the reference to the Paris Agreement (2015)

Line 80: We Explained in the text that $CO_2e$ refers to $CO_2$ equivalents. We also kept throughout the text the $CO_2$-eq format.

Line 81: We added footnote no. 5 explaining the GWP concept. We added as well the reference to IPCC 2014, AR5 report.

Line 81: We added footnote no. 4 explaining that FOLU is the same with LULUCF. For consistency, we refer throughout the text to LULUCF. The main reason for naming FOLU LULUCF is that FOLU represents together with Agriculture (AFOLU) a new sector under IPCC AR5, while countries report under the UNFCCC NGHGI net $CO_2$ emissions and removals from LULUCF. Also, widely used in the literature we see the terminology incl./excl. LULUCF.

Lines 81 and 87: No, there is no difference between the two expressions, we inserted everywhere for consistency the "NGHGI 2018 estimates" throughout the manuscript.

Figure 1: we extended the explanations in a new Caption. We explained as well the concept of remaining and converted land. We added footnote 6 to explain the UNFCCC LULUCF classes.

Line 106: Same as line 81, explained in footnote 4

Line 112 and 120: No, these emissions do not include LULUCF estimates, we added ion brackets (excl. LULUCF). We also added a sentence explaining why we exclude the $CH_4$ emissions from LULUCF.

Line 131: Yes, $CH_4$ and $N_2O$ results from top-downs estimates will be included in the following synthesis. We therefore, as suggested, moved this sentence to Conclusions

Line 152 and Table 2: We added the $CO_2$ emissions from LULUCF to Table 2

Line 154: we replaced "observational data streams" to "modelled and reported data-streams"

Figure 2 legend: we added the following section "The values in this study are defined from an atmospheric point of view, which means that positive values represent a source to the atmosphere and negative ones a removal from the atmosphere." to the introductory paragraph of section 2.

Line 198-200: we moved the $CO_2$ information to the introductory paragraph of section 3.

Line 325: We updated the Figures 3 and 7 by adding as well the difference between 2012 and 2005. 2012 is the last common year where all data sources have estimates. We updated the text and figures discussion accordingly.

Line 386: We replaced "sources" with "estimates". The data belongs to $CH_4$ emissions from wetlands, so yes, are natural $CH_4$ emissions.

Line 466: We updated the DGVM definition as following: *"DGVMs calculate NBP as the net flux between land and atmosphere defined as photosynthesis minus the sum of plant and soil heterotrophic respiration, carbon fluxes from fires, harvest, grazing, land use change and any other C flux in/out of the ecosystem (e.g. DIC, DOC, VOCs)"*

Line 499: If afforestation is removed the sink will decrease, we therefore corrected the statement. For consistency, we changed as well the colors of Figure 10 to match the UNFCCC and FAOSTAT colors used in Figures 2,3,5,6,7,8.

Line 516: We added the footnote 9 explaining what FRA means. The link was as well corrected.

Line 543-546: We rephrased this paragraph: *"Some of these are found in northern Europe (e.g. Finland and Sweden), while others are in the far south, i.e. the south of Spain. In 2015 just above one fifth of the EU28's area (21 %) was covered by grassland. There is a broad range across EU Member States, with Ireland having 56% of its total land area as grassland and Finland and Sweden less than 6 % of the land (EUROSTAT: https://ec.europa.eu/eurostat/statistics-explained/index.php/Land_cover_statistics)."*

Line 557: We added a paragraph stating which countries are triggering the high grassland estimates: *"The high estimates of grassland emissions in 2016 UNFCCC NGHGI submissions are explained by increased emissions in Austria, Denmark, Croatia; Sweden changed from being a sink in 2015 to being a very high source in 2016 and Hungary and Greece reported lower sink. Ireland was the only country which reported a higher sink in 2016 compared to 2015"*

Line 625: Yes, Houghton estimates are the same as H&N estimates, we changed the text accordingly.

Figure 13: We uploaded on the new zenodo link [https://zenodo.org/record/3662371#.Xkui-WhKjIU](https://zenodo.org/record/3662371#.Xkui-WhKjIU) the data for Figure 13. We deleted the term "Model (cumulative 1990)" and updated the names of the data sources and colours for consistency with the other Figures (e.g. Figure 10) in the manuscript.

Line 676: Yes, it is true, we deleted the word "InGOS" and left the reference.

Line 728: Freshwater $CO_2$ is not a disturbance, we rephrased: "carbon lost due to a disturbance (e.g. forest fire, harvest)"

Figure 14 legend: We provide a clearer figure caption

Line 811: We added some sentences at the end of the Conclusions paragraph.

Line 815: total emissions column: refers to the $XCO_2$ = the column-averaged dry-air mole fractions of $CO_2$ and $XCH_4$ = the column-averaged dry-air mole fractions of $CH_4$

Line 815ff: we added the two references as suggested together with a sentence on regional inversions.

Appendix B UNFCCC: the whole description was re-written – this refers also to the remaining remarks

Line 884ff: the new UNFCCC updated text includes a clearer explanation on Monte Carlo simulations and uncertainty approaches according IPCC 2006 guidelines.

Table B1: No sectors are missing; the table explains which sub-sectors are aggregated for uncertainty calculation purposes.

Line 977: we added the following explanation*: The correction factor is used as an empirical adjustment, based on Monte Carlo simulations, to correct for the deviation introduced by using the "standard" uncertainty calculation*

*method suggested by IPCC error propagation which is only a first order approximation; for large uncertainties (as they accumulate in the propagation chain) the method systematically underestimates the uncertainty half range.*

**Response to the technical corrections:**

Line 61: We inserted the explanation of DGVM

Line 87 and Figure 1: We use now consistent $CO_2$-eq everywhere in the paper

Line 129: changed to top-down GHG emission estimates

Table 1: We added a line for each gas to better separate between the three sections.

Table 2: We changed the structure of the table by adding lines in between activities

Figures 2,5,6,10 legends: we kept throughout the whole study UNFCCC NGHGI 2018

Line 345: We replaced the reference with the name of the excel table

Line 345: We rephrase to be clear that "count as much as" refers to the listed countries

Line 346: reported to UNFCCC is correct

Line 394: changed to depend

Line 395: changed to ..estimate to have..

Line 425: added to the list of abbreviations the GHGI acronym

Line 450: changed to separately

Line 451: added comma before because

Line 457: We clarified the sentence structure

Line 576: we rephrased the sentence as following: *"DGVMs estimate net land use emission as the difference between a run with and a run without land-use change, and their estimate includes the loss of additional sink capacity, that is, the sink that favors the environmental changes(e.g. $CO_2$ fertilization).."*

Figure 12: We added explanation to the caption regarding the definition of anthropogenic land in DGVMs. The GCP version is the 2018 (Le Quere et al., 2018)

Line 666: we removed "takes"

Line 685: "According to" added

Line 730: we removed "last"

Line 741: added "the" temperature. This phrasing is in line with the IPCC 1.5°C report (https://report.ipcc.ch/sr15/pdf/sr15_spm_final.pdf) and Paris Agreement: **"**The Paris Agreement sets out a global framework to avoid dangerous climate change by limiting global warming to well below 2°C and pursuing efforts to limit it to 1.5°C. " We added "as set by the PA"

Line 768: we deleted the two stars as they have no meaning

Line 783: changed to country reports

Line 788: it is following not flowing

Line 819: the reference is correct, the JRC report include results from the InGOS project. We deleted the word "InGOS"

Line 899: we intended "uncertainties" we deleted "level"

Line 983: CAP Common Agriculture Policy , we added it to the list of acronyms

We added as well the three references as you kindly suggested.

[revised manuscript text omitted]